# Phase II trial of cytarabine and mitoxantrone with devimistat in acute myeloid leukemia

Rebecca Anderson[1], Lance D. Miller [2], Scott Isom[3], Jeff W. Chou[3], Kristin M. Pladna[1], Nathaniel J. Schramm[1], Leslie R. Ellis[1], Dianna S. Howard[1], Rupali R. Bhave[1], Megan Manuel[1], Sarah Dralle[1], Susan Lyerly[1], Bayard L. Powell[1] & Timothy S. Pardee [1,2,4✉]

Devimistat is a TCA cycle inhibitor. A previously completed phase I study of devimistat in combination with cytarabine and mitoxantrone in patients with relapsed or refractory AML showed promising response rates. Here we report the results of a single arm phase II study (NCT02484391). The primary outcome of feasibility of maintenance devimistat following induction and consolidation with devimistat in combination with high dose cytarabine and mitoxantrone was not met, as maintenance devimistat was only administered in 2 of 21 responders. The secondary outcomes of response (CR + CRi) and median survival were 44% (21/48) and 5.9 months respectively. There were no unexpected toxicities observed. An unplanned, post-hoc analysis of the phase I and II datasets suggests a trend of a dose response in older but not younger patients. RNA sequencing data from patient samples reveals an age-related decline in mitochondrial gene sets. Devimistat impairs ATP synthesis and we find a correlation between mitochondrial membrane potential and sensitivity to chemotherapy. Devimistat also induces mitochondrial reactive oxygen species and turnover consistent with mitophagy. We find that pharmacological or genetic inhibition of mitochondrial fission or autophagy sensitizes cells to devimistat. These findings suggest that an age related decline in mitochondrial quality and autophagy may be associated with response to devimistat however this needs to be confirmed in larger cohorts with proper trial design.

[1] Section on Hematology and Oncology, Comprehensive Cancer Center of Atrium Health Wake Forest Baptist, Winston-Salem, NC, USA. [2] Department of Cancer Biology, Comprehensive Cancer Center of Atrium Health Wake Forest Baptist, Winston-Salem, NC, USA. [3] Department of Biostatistics and Data Science, Wake Forest Public Health Sciences, Winston-Salem, NC, USA. [4] Rafael Pharmaceuticals Inc, Cranbury, NJ, USA. ✉email: tspardee@wakehealth.edu

Acute myeloid leukemia is an aggressive malignancy with poor outcomes especially in patients 60 years of age or older. This is thought to be in part from increased resistance to chemotherapy in AML cells arising in an older host[1]. Treatment with induction chemotherapy results in the attainment of a complete remission (CR) rate of up to 80% in younger patients and 50–60% in older patients[2]. Despite these high remission rates relapse is common and once relapsed the attainment of a second remission is more difficult[2]. Once relapsed the only curative treatment modality is an allogenic stem cell transplant. Patients in remission at the time of transplant have the highest long-term survival rates[3,4]. Therefore, treatments that produce high remission rates are particularly important. There is no standard salvage regimen for patients with relapsed or refractory AML and novel approaches are needed.

Metabolism is altered in cancer cells and attempts to leverage these differences into therapeutic strategies have been extensively studied (reviewed in refs. [5–7]). Unfortunately, the targeting of non-mutated metabolic enzymes has not yet yielded a clinically successful strategy. Devimistat (CPI-613) is a novel lipoate analog that mimics the catalytic intermediates of both pyruvate dehydrogenase (PDH) and α-ketoglutarate dehydrogenase (KGDH) complexes. This results in stimulation of inhibitory regulatory processes causing simultaneous inhibition of both complexes[8,9]. In early clinical trials in pancreatic cancer patients, the addition of devimistat resulted in impressive response rates[10] leading to a phase III clinical trial[11]. Mitochondrial metabolism is a source of resistance in AML. Studies have shown an increased mitochondrial mass and an oxidative-phosphorylation gene expression signature in cytarabine-resistant AML cells[12]. In addition, AML cells display a dose-dependent increase in mitochondrial oxygen consumption when treated with chemotherapy, and this mitochondrial respiration is a source of resistance[13]. In preclinical models, devimistat sensitized AML cells to chemotherapy and decreased mitochondrial respiration leading to a phase I study in relapsed and refractory AML patients[13].

In the previous phase I study, the maximally tolerated dose of devimistat when given in combination with high-dose cytarabine and mitoxantrone was 2500 mg/m². However, there was no clear dose–response relationship observed. A single-arm phase II study was conducted of the combination of devimistat with high-dose cytarabine and mitoxantrone followed by maintenance devimistat for relapsed or refractory AML patients. The primary endpoint for the trial was the feasibility of maintenance devimistat with the secondary endpoints of response, defined as the complete remission plus the complete remission rate with incomplete count recovery rates (CR + CRi), median survival and toxicity profile. In an unplanned, post hoc analysis, the combined datasets from the phase I and II studies were analyzed to determine patient characteristics associated with an increased likelihood of response. In addition to the clinical endpoints of the phase II trial the current report details the results of this unplanned post hoc analysis that found a dose response in older but not younger patients and mechanistic studies showing devimistat induced mitochondrial reactive oxygen species, mitophagy, and a relationship between mitochondrial membrane potential and response to the chemotherapy sensitizing effects of devimistat.

## Results

**Baseline demographics and safety.** The complete phase II protocol is provided in Supplementary Note 1. The phase II patient characteristics are summarized in Supplementary Table 1 and enrollment details are in the CONSORT diagram (Supplementary Fig. 1). Briefly, 47 patients with relapsed or refractory AML and one patient with a relapsed granulocytic sarcoma were enrolled in study. The median age of the cohort was 64 years old. The median marrow blast involvement at the time of enrollment was 30%. Of the 48 patients enrolled, all but one of them received initial cytotoxic chemotherapy induction with 90% receiving 7 + 3 as initial treatment, 81% received study treatment as their first salvage, 13% received previous high-dose cytarabine (HiDAC) based salvage, 27% prior hypomethylating agent and 4% of patients had received prior stem cell transplant. Disease status was refractory in 29% and relapsed in 71% of patients. For patients with a previous remission, the median duration of the first remission was 11.5 months. Cytogenetics were poor in 38%, intermediate in 52%, good in 4%, and unknown in 6%.

All patients who received at least one dose of CPI-613 were included in the safety analysis (n = 48). The treatment schema for the trial is summarized in Supplementary Fig. 2. Thirty-day mortality was 8% (4/48) and 60-day mortality was 27% (13/48). This was similar to phase I with 12% (8/67) and 19% (13/67), respectively. Historical experience with the same HiDAC, mitoxantrone dose, and schedule but with the addition of asparaginase (HAMA) had a 13% 30-day and 22% 60-day mortality[14]. Hematological toxicities were the most common as expected for a HiDAC based regimen with all patients having grade 3–4 hematologic toxicities. Diarrhea, mainly grade 1 and 2, was the next most common toxicity with 89% of patients affected. Infectious complications were also common as expected in this patient population with 50% of patients having at least one episode of neutropenic fever. The first seven patients were treated with 2500 mg/m² of devimistat. After increased rates of diarrhea were seen, the protocol was amended to discontinue this dose and the remaining patients were treated with either 2000 mg/m² (n = 20) or 1500 mg/m² (n = 21). Toxicities by the dose of devimistat are listed in Supplementary Table 2, all toxicities are reported by grade and patients may have experienced related toxicities captured under the same heading. For this reason, many headings have more instances than patients treated. All percentage of affected patients calculations were done by the number treated at that dose.

**Primary and secondary outcomes of the phase II study.** The primary objective of the phase II study was to determine if the maintenance schedule of devimistat was feasible. This was defined as 50% of eligible patients completing three cycles of maintenance. Of the 21 patients who achieved a remission and were therefore eligible for maintenance, 19 did not receive maintenance therapy. One was lost to follow-up, three did not receive maintenance because of physician choice, five relapsed during consolidation and ten went on to allogeneic stem cell transplant. Of the two patients who started maintenance therapy one refused all consolidation therapy while the other completed two cycles of consolidation. Both patients relapsed prior to cycle 3 of maintenance. Therefore, the primary endpoint of the study was not met as none of the eligible patients completed three cycles of maintenance therapy. For the secondary outcome of response, all of the 48 patients were evaluable for response. The composite response rate was 44% with 15 complete remissions (CR) and 6 complete remissions with incomplete count recovery (CRi). For the secondary outcome of overall survival with a median follow-up of 38 months, median survival for all 48 patients was 5.9 months.

**Unplanned post hoc analysis reveals a dose response of devimistat in older but not younger patients.** Previous analysis of the activity of devimistat in combination with high-dose cytarabine (HiDAC) and mitoxantrone suggested that older and younger patients had similar outcomes in contrast to essentially all previous

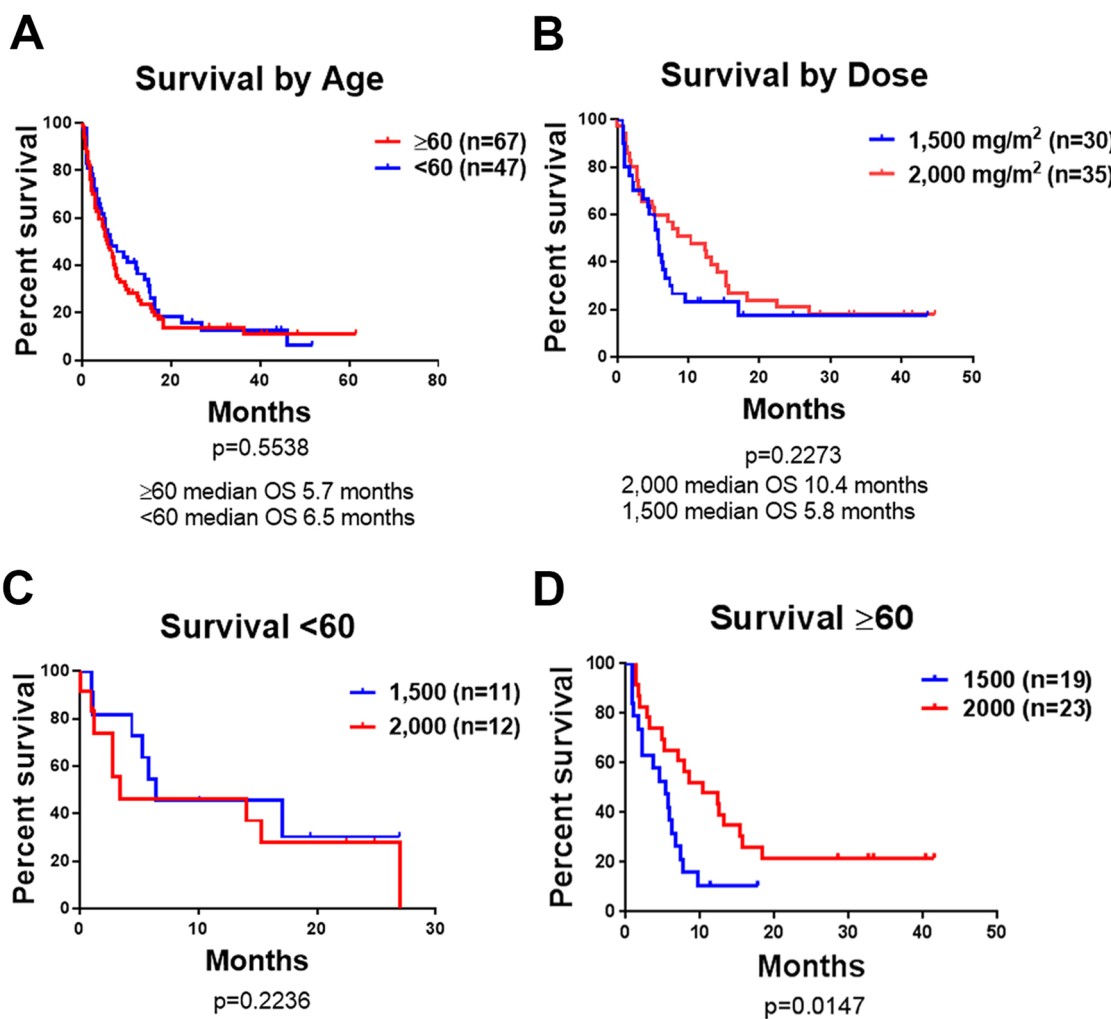

**Fig. 1 Efficacy of HiDAC, mitoxantrone, and devimistat. A** Overall survival of the combined phase I and II cohorts by age. **B** Overall survival of the combined phase I and II cohorts by dose of devimistat. **C** Overall survival of the combined phase I and II cohorts by dose of devimistat for patients <60 years of age. **D** Overall survival of the combined phase I and II cohorts by the dose of devimistat for patients ≥60 years of age. Log-rank test was used to assess the difference between survival curves, *P* value less than 0.05 was considered significant.

studies utilizing a cytotoxic backbone in AML[15]. To confirm this observation, an unplanned post hoc analysis of the combined data from the phases I and II for all 114 AML patients treated were completed. Median survival in all patients treated with devimistat was assessed by age. The Kaplan–Meier curves for both age groups completely overlap with a median OS of 6.5 months in younger patients compared to 5.7 months in patients 60 years of age or older (*P* = 0.5538, Fig. 1A). This data suggests a differential benefit of the addition of devimistat in older patients. In order to assess the comparative activity of the 1500 and 2000 mg/m² regimens the median survival of all patients treated with either dose of devimistat was analyzed. The combined demographics of these cohorts are summarized in Table 1. There was a numerical advantage for the 2000 mg/m² dose with a median survival of 10.4 versus 5.8 months although it was not significant (*P* = 0.2273, Fig. 1B). When these cohorts were analyzed by age, patients younger than 60 showed no significant difference in survival between doses (Fig. 1C). While the number of patients over 60 was modest (23 treated with 2000 mg/m² and 19 treated with 1500 mg/m²), the two dose cohorts were well matched in median age, percentage of refractory patients, and cytogenetic risk. Despite the small numbers, there was a significant survival advantage for patients over 60 treated with the 2000 mg/m² devimistat dose compared to those treated with the 1500 mg/m² dose with a median survival of 10.4 versus 5.4 months (*P* = 0.0147,

Fig. 1D). Response rates were also increased with 52% (12/23) of patients achieving a CR or CRi in the 2000 mg/m² cohort compared to 37% (7/19) in the 1500 mg/m² cohort (Table 2).

**RNA sequencing analysis.** In order to assess underlying biological differences between older and younger patients, an RNA sequencing analysis of all available patient samples was conducted. There were 17 baseline bone marrow samples from the phase I study and an additional 8 samples from the phase II study with sufficient quality RNA for analysis. Since devimistat targets mitochondrial metabolism and mitochondria can be transferred from marrow stromal cells to AML cells when treated with chemotherapy[16,17] RNA from all mononuclear cells in the bone marrow were included in the analysis. Gene set enrichment analysis (GSEA) is a computational method that assess the degree of concordant expression of sets of genes in common biological pathways as a function of a predefined condition[18]. When the RNA sequencing expression data of the samples was analyzed for gene sets with expression changes correlated to age there was a highly significant negative association between age and gene sets involved in mitochondrial biology. These included genes involved in cellular respiration, oxidative phosphorylation, electron transport chain, ATP synthesis coupled to electron transport, and mitochondrial respiratory chain complex

**Table 1 Demographics of 2000 and 1500 mg/m$^2$ cohorts.**

| Age | All | All | P value | ≥60 y/o | ≥60 y/o | P value |
|---|---|---|---|---|---|---|
| Dose of devimistat (CPI-613®) (mg/m$^2$) | 2000 ($n = 35$) | 1500 ($n = 30$) | | 2000 ($n = 23$) | 1500 ($n = 19$) | |
| Demographics | | | | | | |
| Male | 57% (20/35) | 53% (16/30) | 0.8061 | 70% (16/23) | 68% (13/19) | 1.0000 |
| Median age (range) | 63 (21–76) | 63 (25–80) | 0.7371 | 69 (60–76) | 68 (60–80) | 0.4325 |
| Line of salvage (range) | 1 (1–5) | 1 (1–3) | 0.3646 | 1 (1–2) | 1 (1–3) | 0.0352 |
| Refractory to initial therapy | 26% (9/35) | 37% (11/30) | 0.4224 | 26% (6/23) | 21% (4/19) | 1.0000 |
| Duration of CR1 (months) | 9.5 | 8 | 0.7126 | 11 | 7 | 0.0690 |
| Cytogenetic risk | | | 0.1267 | | | 0.5863 |
| Poor | 43% (15/35) | 43% (13/30) | | 43% (10/23) | 42% (8/19) | |
| Intermediate | 49% (17/35) | 47% (14/30) | | 48% (11/23) | 49% (10/19) | |
| Good | 9% (3/35) | 0% (0/30) | | 9% (2/23) | 0% (0/19) | |
| Unknown/NA | 0% (0/35) | 10% (3/30) | | 0% (0/23) | 5% (1/19) | |
| Outcomes | | | | | | |
| Median survival (months) | 10.4 | 5.8 | 0.2273 | 10.4 | 5.4 | 0.0147 |
| Response (CR + CRi) | 43% (15/35) | 50% (15/30) | | 52% (12/23) | 37% (7/19) | |
| Went to transplant | 26% (9/35) | 23% (7/30) | | 22% (5/23) | 11% (2/19) | |

**Table 2 Responses 60 years of age and older.**

| Devimistat dose | 1500 mg/m$^2$ devimistat ($n = 19$) | 2000 mg/m$^2$ devimistat ($n = 23$) |
|---|---|---|
| Median survival (months) | 5.4 | 10.4 |
| Response (CR + CRi) | 37% (7/19) | 52% (12/23) |
| Went on to transplant | 11% (2/19) | 22% (5/23) |

assembly (Fig. 2A–E). Furthermore, this association was verified in a separate AML cohort with RNA sequencing data corresponding to 239 baseline bone marrow specimens annotated with patient age[19] (Fig. 2F). These findings demonstrate a significant age-related decline in the expression of genes involved in mitochondrial function and assembly. To confirm and extend this result, Western blots of primary patient samples from several patients of different ages for members of the electron transport chain (ETC) was performed. Consistent with the RNAseq data there was a decrease in expression of ETC components in the older compared to younger samples (Supplementary Fig. 3A, B). To further address if the age of the hematopoietic stem cell (HSCs) that gives rise to AML effects mitochondrial function, HSCs from either fetal livers (Young) or retired breeders (Old) were transduced with MLL-ENL fusion gene and NRas$^{G12D}$ and the resulting AML grown out. The Young AML cell line demonstrated increased gene expression of several mitochondrial-associated genes, a significantly increased basal oxygen consumption rate (OCR), more mitochondria, and a small but significant increase in mitochondrial membrane potential (MMP) (Supplementary Fig. 3C–E). To further extend these results MLL-ENL and NRas$^{G12D}$ leukemias were generated from HSCs from adolescent mice (6 weeks of age) or middle-aged mice (15 months of age) and mitochondrial content and MMP were assessed. Consistent with the previous data the adolescent-derived leukemia had a higher mitochondrial content and MMP (Supplementary Fig. 3F). These data complement the expression data and support the hypothesis that AML arising from older HSCs have a less functional mitochondrial compartment.

**Mitochondrial membrane potential and devimistat.** The above data suggest that increased benefit from devimistat in older patients may be related to impaired mitochondrial function. MMP is an essential driver of mitochondrial ATP generating capacity and is predicted to be impaired based on the gene expression pathways that decreased with age and supported by the functional studies of AML arising from older HSCs (Supplementary Fig. 3). Devimistat inhibits mitochondrial respiration in leukemia cells[20] but a direct effect on ATP production has not been established. To address this, AML cells were incubated with increasing amounts of devimistat for 6 h, and viability and ATP concentrations determined. ATP levels fell significantly upon incubation with devimistat (Fig. 3A). This suggests that AML cells with increased MMP and ATP generating capacity would be more resistant to devimistat. In order to study the relationship between MMP and devimistat response three genetically defined mouse models of AML with different baseline MMP were utilized (Fig. 3B). The MFL2 cell line with the highest MMP had the least response to the addition of devimistat to doxorubicin (Fig. 3C). RN2 cells were in between in both measures (Fig. 3D) and RHRAS cells with the lowest MMP had the largest effect (Fig. 3E). These data suggest a correlation between MMP and effect of devimistat; however, to establish causality, the cell line with the highest MMP (MFL2) was treated with the clinically available complex one inhibitor metformin. Cells treated with metformin demonstrated a significant decline in MMP (Fig. 3F). The addition of metformin to doxorubicin and devimistat resulted in a significant increase in response compared to devimistat and doxorubicin alone (Fig. 3G). These data suggest that MMP is a source of resistance to devimistat. The ability of metformin to sensitize MFL2 cells to devimistat suggested the combination maybe synergistic. A combinatorial index (CI) is a mathematical model that quantifies the degree of synergy between two agents[21] where CI values below 1 are considered moderately synergistic and those below 0.7 are considered synergistic[22]. There was significant synergy between devimistat and metformin with a CI value of 0.67 (Fig. 3H). In addition to its ETC inhibitory effects, metformin was recently shown to induce reactive oxygen species (ROS) as part of its anticancer activity[23,24]. To determine if metformin-induced ROS played a role in the observed synergy, we exposed AML cells to devimistat, metformin, or both and assessed mitochondrial ROS. Consistent with previous reports devimistat significantly induced mitochondrial ROS[8] (Supplementary Fig. 4A). Metformin alone did not significantly increase mitochondrial ROS however; the combination of devimistat and metformin did increase mitochondrial ROS above devimistat alone (Supplementary Fig. 4A). To determine if this excess ROS contributed to efficacy, AML cells were exposed to devimistat with and without metformin in the presence of the ROS scavenger n-acetyl-cysteine (NAC). There was no attenuation of the

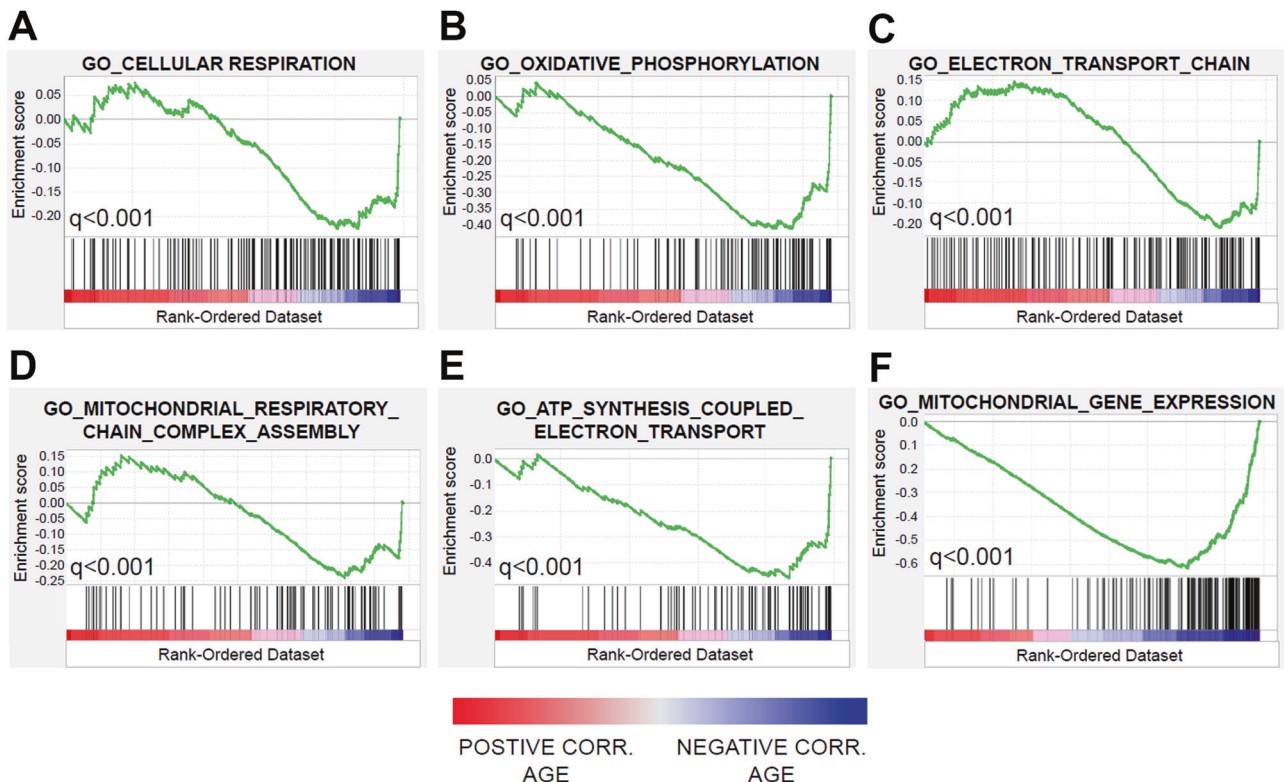

**Fig. 2 Genes inversely correlated with patient age are enriched for mitochondrial functions. A–E** RNA sequence data for 25 baseline bone marrow samples from the combined phase I + II cohorts were analyzed for gene expression patterns correlated with patient age. Genes were ranked on Pearson correlation coefficients and analyzed using the Gene Set Enrichment Analysis (GSEA) tool kit and Molecular Signatures Database. Shown are the top most enriched gene sets negatively associated with age. No significant enrichment was observed for positively correlated genes. **F** The most significant age-gene expression gene set from the OHSU AML RNAseq dataset of 239 baseline bone marrows is shown, confirming the negative association between mitochondrial AML gene expression and patient age. In this dataset, the gene sets represented in panels **A**–**E** also showed significant enrichment ($q < 0.004$ for all gene sets). Enrichment scores (green line) and their corresponding false discovery rate-adjusted significance ($q$ values) are shown. Black vertical bars indicate the relative position of genes belonging to the gene set along the rank-ordered dataset. GO Gene Ontology GO terms.

cytotoxicity in the presence of NAC, in fact the addition NAC resulted in numerically lower viability though this did not reach significance (Supplementary Fig. 4B). To confirm that NAC was attenuating ROS AML cells were incubated with devimistat, metformin, NAC, or the combination and mitochondrial ROS assessed. Surprisingly, NAC when added to devimistat and metformin actually significantly increased mitochondrial ROS compared to devimistat and metformin alone (Supplementary Fig. 4C). To assess the ability of NAC to attenuate mitochondrial ROS not related to devimistat AML cells were treated with the protonophore Carbonyl cyanide-*p*-trifluoromethoxyphenyl hydrazone (FCCP) or devimistat with and without NAC and mitochondrial ROS assessed. Consistent with the previous result NAC significantly increased mitochondrial ROS in the presence of devimistat but significantly attenuated it with FCCP (Supplementary Fig. 4D). This demonstrates the failure of NAC to rescue devimistat- and metformin-treated cells is not related to decreased mitochondrial ROS. To more directly test the relative roles of ETC activity and mitochondrial ROS in the activity of devimistat, AML cells were treated with devimistat with the complex III inhibitor antimycin A or the mitochondrial ROS-generating herbicide paraquat. Both antimycin A and paraquat significantly sensitized AML cells to devimistat (Supplementary Fig. 4E, F) suggesting that both ETC inhibition and ROS generation can sensitize cells to devimistat. In addition, metformin did not significantly increase the sensitivity of the RHRAS cell line with the lowest MMP supporting an effect on ETC (Supplementary Fig. 4G). As metformin is an approved agent for the

treatment of diabetes allowing rapid translation, the in vivo activity of this combination was determined. Syngeneic C57Bl/6 mice were injected with MFL2 cells and upon engraftment treated with PBS, devimistat alone, metformin alone or combined with devimistat. Combination-treated mice had a modest but highly significant survival benefit when compared to controls or either treatment alone (Fig. 3I). This survival benefit was similar in magnitude to the benefit of cytarabine and doxorubicin in previously published studies of this aggressive model[25]. To confirm and extend these results, devimistat and metformin were tested alone and in combination in a PDX model derived from a 75-year-old patient. Consistent with the results from the syngeneic model there was a modest but significant benefit of the combination compared to either single agent alone (Supplementary Fig. 4H). These data suggest that MMP is a source of resistance and mitochondrial ROS potentiates the effects of devimistat.

**TCA cycle inhibition leads to increased mitochondrial turnover.** The GSEA revealed electron transport chain assembly as significantly negatively associated with age (Fig. 2E) suggesting that mitochondrial turnover may contribute to the response to devimistat. To determine if devimistat causes increased turnover of mitochondria, treated MFL2 cells were lysed and cytosolic and mitochondrial fractions were blotted for actin, and the mitochondrial proteins VDAC1 and TOM20 (Fig. 4A). Devimistat treatment resulted in a decrease of VDAC1 and TOM20 consistent with increased mitochondrial turnover. To determine if

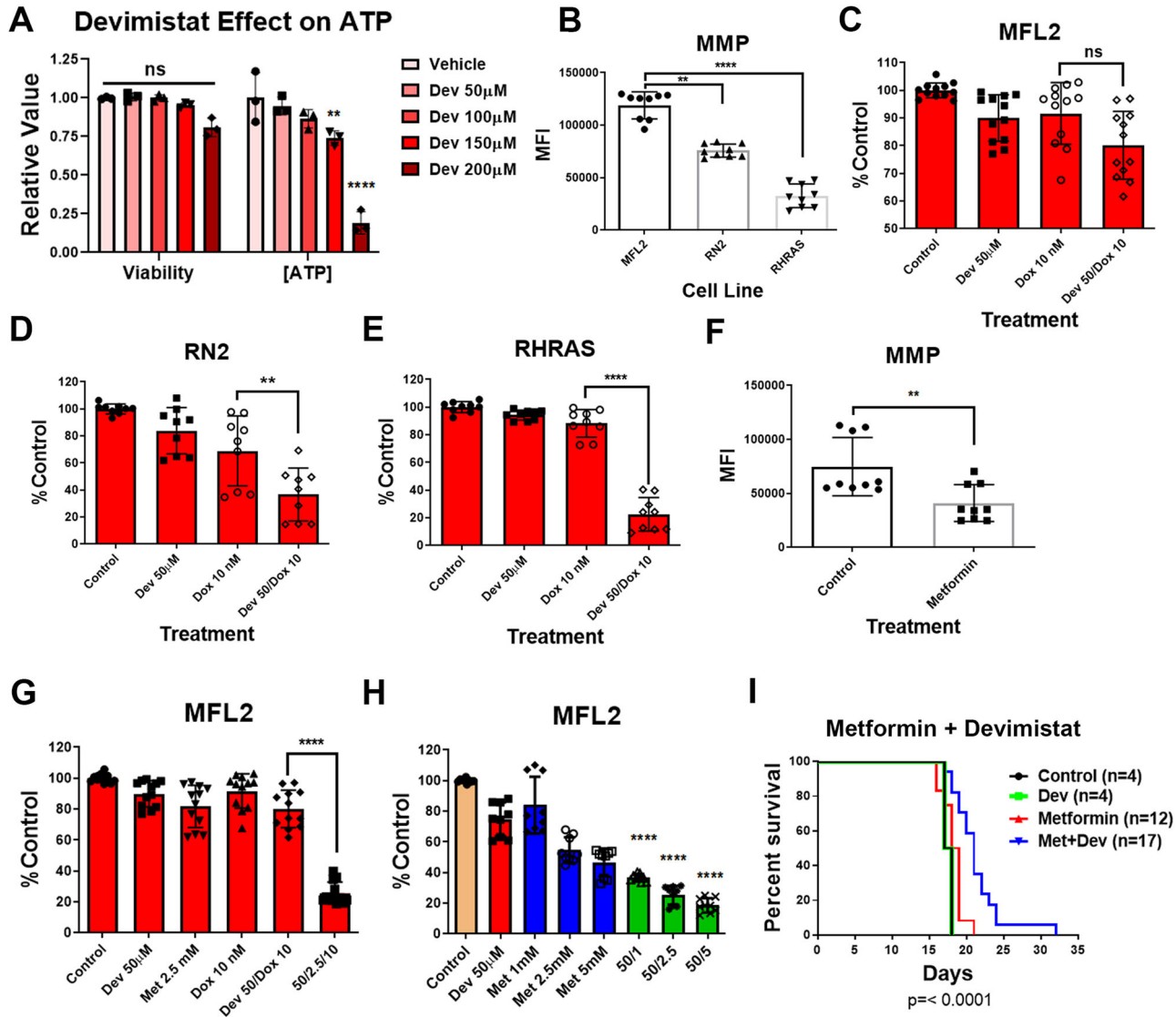

**Fig. 3 Activity of devimistat depends on baseline MMP. A** ATP concentrations. MFL2 cells were incubated with the indicated amount of devimistat for 6 h and assayed for viability using trypan blue exclusion and ATP content using Cell Titer Glo. Means from three independent experiments are shown. **B** TMRM staining. Three genetically defined murine AML cell lines were stained with TMRM and assessed by flow cytometry. Median fluorescence intensity of TMRM stained cells from three independent experiments were quantitated and plotted. **C–E** Three genetically defined murine AML cell lines were exposed to the indicated chemotherapy, devimistat (CPI-613) or both for 72 h and assessed for viability. Shown are the means of at least three biological replicates each done in triplicate. **F** MFL2 cells untreated (control) or treated with 2.5 mM metformin for 24 h were stained with TMRM and assessed by flow cytometry. Median fluorescence intensity of TMRM stained cells from three independent experiments, each in triplicate were quantitated and plotted. **G** MFL2 cells were exposed to the indicated chemotherapy, devimistat (CPI-613), metformin or combinations for 72 h and assessed for viability. Shown are the means of four biological replicates each done in triplicate. **H** MFL2 cells were exposed to the indicated doses of devimistat (CPI-613), metformin or the combination for 72 h and assessed for viability. Shown are the means of three biological replicates each done in triplicate. **I** Survival of syngeneic C57Bl/6 mice injected with $1 \times 10^6$ MFL2 cells and treated with PBS (Control), metformin, or devimistat or the combination. *P* value calculated by log-rank test. Viability data for panel C were derived from four independent experiments each done in triplicate. All other viability data were derived from three independent experiments each done in triplicate. \*\**P* < 0.01, \*\*\*\**P* < 0.001. All error bars represent the standard deviation around the mean.

this effect is specific for TCA cycle inhibition, MFL2 cells deleted for the E1α subunit of PDH (*Pdha1*) by CRISPR/Cas9[13] were compared to control cells with a *Rosa26* targeted sgRNA. *Pdha1*-deleted cells had significantly lower levels of VDAC1 and TOM20 consistent with TCA cycle defects resulting in increased mitochondrial turnover (Fig. 4B). To determine if the decrease in mitochondrial-related proteins was accompanied by an increase in mitochondrial biogenesis, MFL2 cells were treated with devimistat and expression of several genes involved in mitochondrial biogenesis assessed. Following devimistat treatment, significant

increases in expression were seen with *Tfam, Erra, Errg,* and *Nrf1* (Fig. 4C). Consistent with this being a consequence of TCA cycle inhibition *Pdha1*-deleted cells also displayed increased expression of *Tfam, Erra,* and *Nrf1* compared to isogenic controls (Fig. 4D). Finally, if mitochondrial biogenesis is required for resistance to devimistat then doxycycline, an inhibitor of mitochondrial protein translation[26,27], should sensitize AML cells to its effects. Consistent with this when human or murine AML cells were treated with doxycycline they became sensitized to devimistat (Fig. 4E, F). To rule out the effects of doxycycline on MMP,

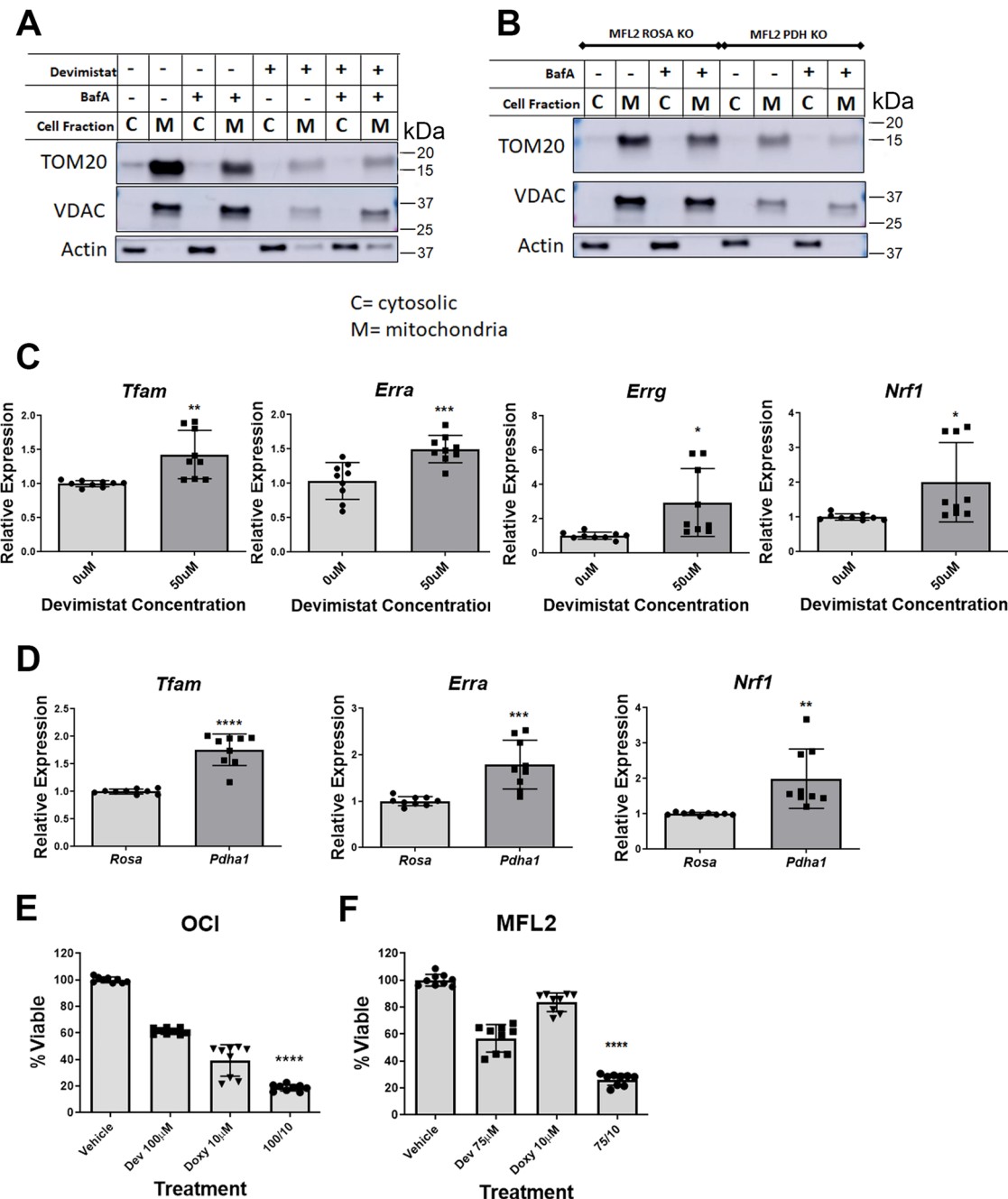

**Fig. 4 Devimistat induces mitochondrial turnover. A** Western blot for Vdac1 and Tom20. MFL2 cells were treated with 100 µM devimistat for 24 h, lysed and cytosolic and mitochondrial fractions blotted for the indicated protein. Actin was used as a localization and loading control. Bafilomycin A (BafA) is an autophagy inhibitor. **B** Control compared to *Pdha1*-deleted cells. Indicated cells were lysed and separated into cytosolic and mitochondrial fractions and blotted for Vdac1 and Tom20. Actin was used as a localization and loading control. **C** QPCR of mitochondrial biogenesis genes. MFL2 cells were treated with the indicated amount of devimistat for 24 h and RNA isolated, converted to cDNA and QPCR performed. Shown are the means of three biological replicates each done in triplicate. Means were compared by two-tailed Students *t* test. **D** QPCR of mitochondrial biogenesis genes. *Pdha1*-deleted and control Rosa26-deleted cells (Rosa) were grown and RNA isolated, converted to cDNA and QPCR performed. Shown are the means of three biological replicates each done in triplicate. Means were compared by two-tailed Students *t* test. **E, F** Viability assays of human (**E**) or murine (**F**) AML cells treated with devimistat (dev) or doxycycline (doxy) or both for 72 h Shown are the means of three biological replicates each done in triplicate. Means were compared by one-sided ANOVA with a Tukey's posttest for multiple comparisons ****$P$ value less than 0.0001. ***$P$ value of 0.001 or less. **$P$ value of 0.01 or less. *$P$ value of 0.05 or less. All error bars represent the standard deviation around the mean.

murine AML cells were treated with doxycycline, devimistat or both for 72 h and TMRM measured. There was no significant decrease in TMRM in cells treated with doxycycline (Supplementary Fig. 5). These data suggest that TCA cycle inhibition in AML cells results in increased mitochondrial turnover.

**Autophagy and devimistat.** The main mechanism of mitochondrial turnover utilizes mitochondrial fission followed by autophagic digestion in a pathway termed mitophagy. Mitochondrial fission requires Drp1 to convert larger mitochondria into smaller fragmented ones. To determine the importance of

mitochondrial fission to TCA cycle inhibition, sensitivity to the Drp1inhibitor MDIVI-1 was assessed. *Pdha1*-deleted cells were significantly more sensitive to MDIVI-1 treatment compared to the *Rosa26* controls (Supplementary Fig. 6A). This increased dependence on mitochondrial fission was also seen in devimistat-treated AML cells since they were more sensitive to MDIVI-1 (Supplementary Fig. 6B). To extend these results, AML cells derived from primary patient samples were treated with devimistat and MDIVI-1 and assessed for viability. Consistent with the cell line models three of the four tested primary patient AML cells were significantly more sensitive to the combination than either agent alone (Supplementary Fig. 6C). Having established AML cells have increased reliance on Drp1 mediated mitochondrial fission in response to TCA cycle inhibition, we next examined autophagy. To determine if devimistat induces autophagy, levels of the autophagic protein LC3-II, which helps to assemble the autophagosome, were monitored in leukemia cells treated with devimistat. Leukemia cells exposed to devimistat showed an accumulation of LC3-II indicative of autophagy induction (Fig. 5A). In order to assess reliance on autophagy in response to TCA cycle inhibition, *Pdha1*-deleted cells and *Rosa26* control cells were treated with the autophagy inhibitor chloroquine. Both mouse (Fig. 5B) and human (Supplementary Fig. 7A, B) *Pdh1a*-deleted cells showed increased sensitivity to chloroquine treatment compared to controls. In addition, AML cells treated with devimistat were more sensitive to chloroquine (Fig. 5C, D). As chloroquine has multiple effects beyond inhibition of autophagy[28] a genetic approach was used to confirm the effect was specific to autophagy. The gene *ATG5* is involved in the elongation and closure of autophagic vesicles[29]. Cells lacking *ATG5* are impaired in classical autophagy although independent autophagic mechanisms remain[30]. AML cells lacking the autophagy gene *Atg5* were generated and assessed for sensitivity to devimistat. Two independent clones deleted for *Atg5* demonstrated an increased sensitivity to devimistat compared to control cells (Fig. 5E, F). In addition, when *Atg5* deleted cells were treated with devimistat TOM20 and VDAC protein expression remained constant in contrast to autophagy competent cells (Fig. 5G and Supplementary Fig. 7C). This finding of preserved TOM20 and VDAC protein levels in cells that have a genetic impairment to autophagy is an orthotopic demonstration of the importance of autophagy in the devimistat induced mitochondrial turnover and further supports that devimistat induces mitophagy. Finally, primary patient-derived AML cells were treated with devimistat and chloroquine and assessed for viability. Primary patient samples were significantly more sensitive to the combination than either agent alone (Fig. 5H). These data taken together demonstrate that TCA cycle inhibition with devimistat induces autophagy-dependent mitochondrial turnover and that inhibition of either autophagy or mitochondrial fission sensitizes cells to devimistat.

**Mitochondrial Turnover in devimistat response is independent of PINK1 and Parkin**. PINK1 and Parkin mediate the main pathway of removal of damage mitochondria. To examine if PINK and Parkin were involved in the devimistat-mediated mitochondrial turnover, AML cells expressing Parkin-mCherry fusion protein were treated with devimistat and stained with mitotracker-deep red. Confocal images revealed that devimistat-treated cells did not demonstrate punctate Parkin that co-localized to the mitochondria (Fig. 6A). In contrast, FCCP-treated cells did show punctate Parkin that co-localized with mitochondria consistent with this pathway being active (Fig. 6A). Since FCCP triggers mitophagy via loss of MMP we determined the effect of devimistat on MMP. AML cells treated with devimistat

showed no decline in MMP up to 24 h post treatment (when mitochondrial turnover was detected, Fig. 6B) suggesting that changes in MMP are not responsible for devimistat induced mitochondrial turnover. As devimistat induces mitochondrial reactive oxygen species and mitochondrial ROS can trigger mitophagy[31] we further examined mitochondrial ROS in devimistat-treated AML cells. Confirming the data in Supplementary Fig. 4, devimistat significantly increased mitochondrial ROS in AML cells. In fact, devimistat induced mitochondrial ROS was far greater than that induced by 2 mM hydrogen peroxide (Fig. 6C). To extend this result, primary patient samples were treated with increasing amounts of devimistat and mitochondrial ROS assessed. As was seen in the cell line models, devimistat induced a robust increase in mitochondrial ROS (Fig. 6D). Intriguingly, the amount of mitochondrial ROS induced by devimistat increased with increasing age. Interestingly, mitochondrial turnover could be partially rescued by NAC, despite its effect of increasing mitochondrial ROS (Fig. 6E). Taken together, our data support a model where devimistat triggers mitochondrial ROS and ultimately mitophagy in the PINK1/Parkin-independent pathway.

## Discussion

The phase II study of devimistat in combination with mitoxantrone and HDAC in relapsed or refractory AML patients did not meet its primary endpoint of demonstrating the feasibility of maintenance devimistat. This was in part secondary to 48% (10/21) of responding patients leaving the study to pursue an allogeneic stem cell transplant. The secondary endpoints of overall remission rate and median survival at 44% and 5.9 months are consistent with other salvage regimens used in this population[32–34]. An important limitation to our study is that the age-related analysis was done post hoc and should therefore be considered hypothesis-generating only. In addition, the phase I and II studies were single-arm studies conducted at a single institution. The effect of the addition of devimistat to chemotherapy in older patients was assessed in the multicenter, randomized phase III trial, ARMADA 2000[35]. Unfortunately, this trial was recently closed for futility. A detailed examination of the data from the trial is not yet available. The data presented here suggest that assessing differences in mitochondrial and autophagic gene expression from responders and non-responders will be crucial in determining if a subset of patients can be found that benefit.

AML is a disease of the elderly with a median onset of 68 years of age. Age is an important prognostic factor in AML with patients 60 years old or older having a much worse prognosis[36,37]. This could be a reflection of the fact that normal hematopoietic stem cells (HSCs) must survive for the entire human lifespan to provide the 200+ billion blood cells that are produced every day[38]. Over time, repeated exposures to environmental carcinogens can select for those HSCs that are most resistant to DNA damage. One of the hallmarks of aging is the progressive loss of mitochondrial function[39]. The dose–response seen in older patients with devimistat may reflect this age-related decline in mitochondria quality and function. The respiratory chain of the mitochondria becomes less efficient as organisms, including humans, age[40]. AML arising in HSCs with declining mitochondria would have limited mitochondrial reserve and be more responsive to the combination of devimistat and chemotherapy. This is supported by the RNA sequence analysis showing a highly significant age-related decline in genes associated with mitochondria and the functional assays from AML induced in younger and older murine HSCs (Supplementary Fig. 3). Consistent with this, AML cell lines with lower MMP were

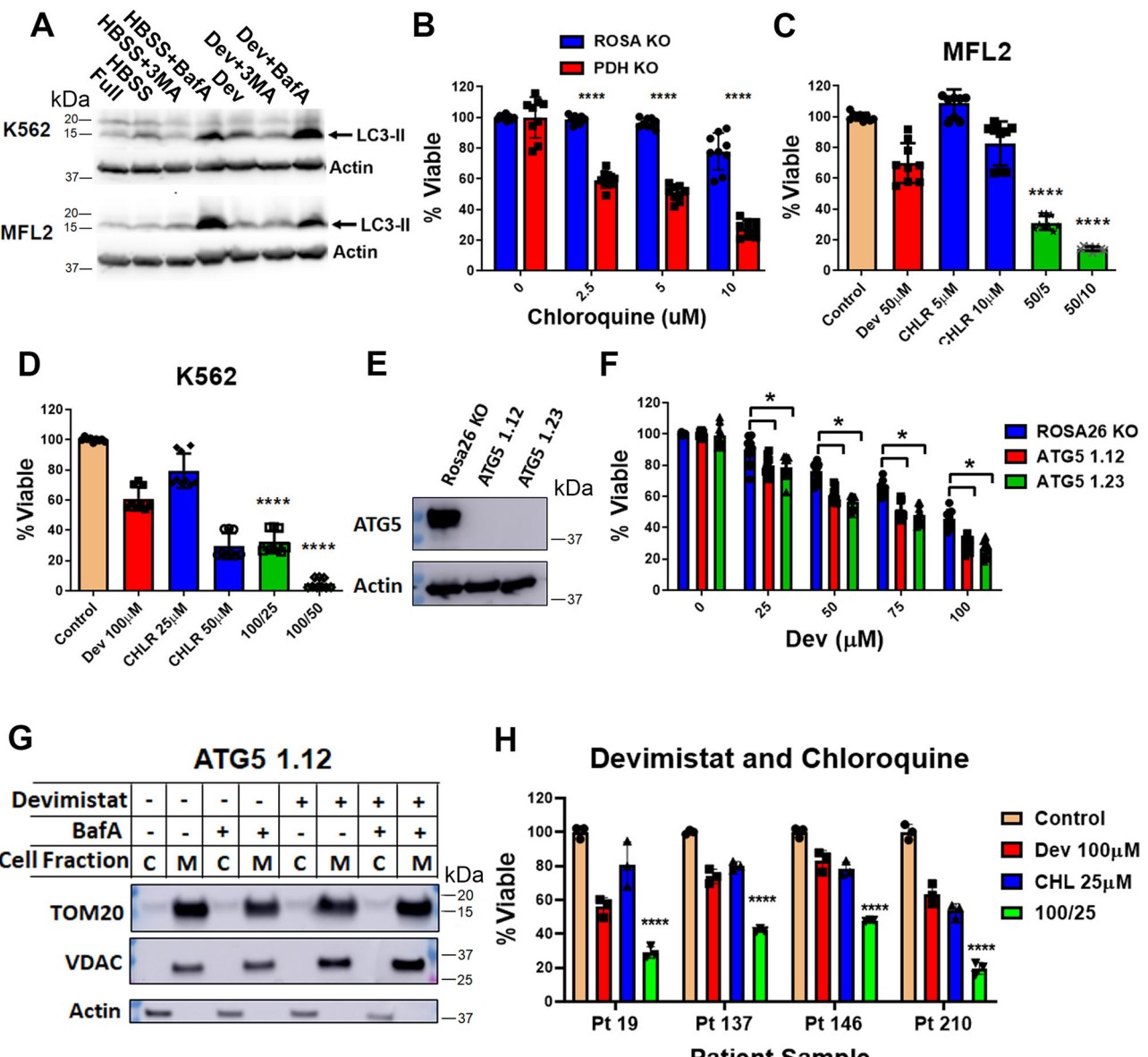

**Fig. 5 Devimistat and mitophagy. A** Western blot for the autophagy protein LC3B-II (LC3-II). K562 or MFL2 cells were incubated for 4 h in normal media (Full) or Hank's balanced salt solution (HBSS) as negative and positive controls or in normal media with devimistat (Dev). Cells were lysed and blotted for LC3B-II. 3MA 3-methyladenine, BafA Bafilomycin A. **B** Viability assay. Pdha1-deleted or Rosa control cells were treated with the indicated amount of chloroquine for 72 h and viability assessed. **C**, **D** Viability assay. K562 or MFL2 cells were incubated with devimistat (CPI), chloroquine (CHLR), or both at the indicated μM concentrations for 72 h and viability assessed. **E** Western blot. MFL2 cells expressing Cas9 were infected with sgRNAs targeting *Atg5* or the safe harbor locus *Rosa26*. Clonal isolates were obtained by serial dilution and blotted for Atg5. Actin is used as a loading control. **F** Viability assays. Clones from (**E**) were incubated in the indicated concentrations of devimistat (CPI) for 72 h and viability assessed. **G** Western blot for Vdac1 and Tom20. *Atg5* KO cells were treated with vehicle or 100 μM devimistat as indicated for 24 h, lysed and cytosolic and mitochondrial fractions blotted for the indicated protein. Actin was used as a localization and loading control. **H** Apoptosis assay. Primary patient sample-derived AML cells were placed in culture and treated as indicated for 72 h. Following treatment cells were stained with annexin V and PI and analyzed by flow cytometry. Each treatment was done in triplicate. For all viability assays, a mean of three independent experiments each done in triplicate are shown. Means compared by two-way ANOVA and multiple comparisons corrected by Sidak's posttest. ****$P < 0.001$, *$P < 0.05$. All error bars represent the standard deviation around the mean.

significantly more sensitive to the combination of devimistat and chemotherapy. This observation is also consistent with several previous studies showing chemotherapy resistance is driven by mitochondrial function[12,41].

The clinically available complex I inhibitor metformin lowered the MMP of a resistant cell line making it sensitive to devimistat and chemotherapy. Further, metformin and devimistat alone provided a survival advantage compared to chemotherapy in an aggressive AML syngeneic model. Metformin has previously been

shown to sensitize AML cells to chemotherapy in preclinical models[42,43], however, clinical benefit in AML patients taking metformin was not seen[44], suggesting additional metabolic stress may be needed. The pairing of TCA cycle inhibition with complex I inhibition may represent such a strategy.

A significant negative correlation was seen in RNA sequence data of patient bone marrow with genes involved in electron transport assembly implicating mitochondrial turnover as a possible resistance mechanism. Cells with impaired TCA cycle

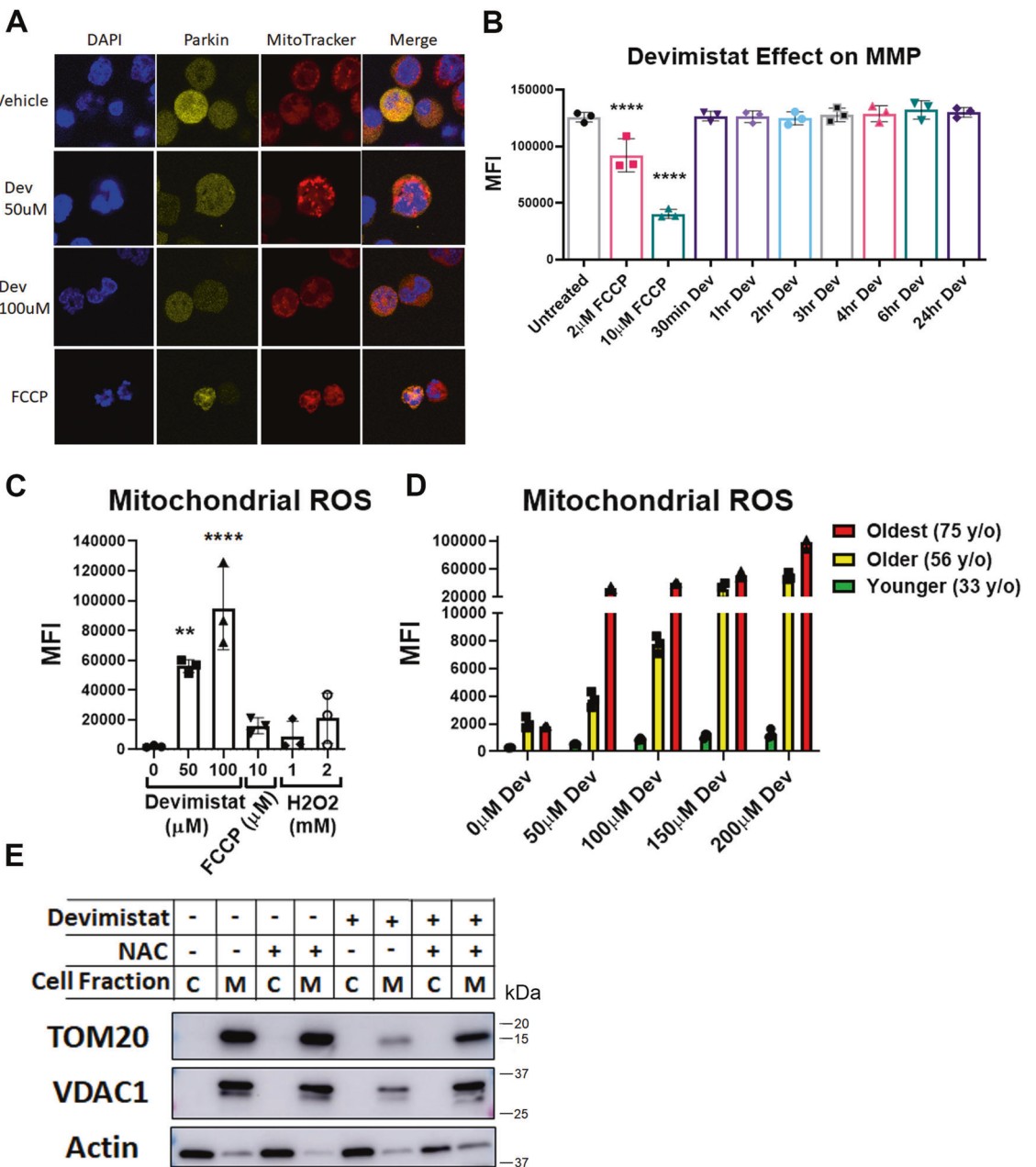

**Fig. 6 Devimistat induces mitophagy via mitochondrial ROS. A** Immunofluorescence. MFL2 cells expressing mCherry-Parkin were treated as indicated for 24 h and imaged using the 60x objective, total magnification of ×600. **B** MMP assessments. MFL2 cells were treated with FCCP for 6 h or 100 μM devimistat for the indicated time and stained with TMRM. Means of three independent experiments are shown. **C** Mitochondrial ROS. MFL2 cells were treated as indicated for 24 h and stained with MitoSOX red to assess mitochondrial ROS. Means of three independent experiments are shown. **D** Mitochondrial ROS in primary patient-derived AML cells. Primary patient sample-derived AML cells were placed in culture and treated as indicated for 24 h. Cells were then stained with MitoSOX red to assess mitochondrial ROS. Shown are the means of triplicates. **E** Western blot for Vdac1 and Tom20. MFL2 cells were treated with vehicle, 100 μM devimistat with or without n-acetyl-cysteine (NAC) as indicated for 24 h, lysed and cytosolic and mitochondrial fractions blotted for the indicated protein. Actin was used as a localization and loading control. For (**B**–**D**), a mean of three independent experiments are shown. Means compared by two-way ANOVA and multiple comparisons corrected by Sidak's posttest. *$P < 0.05$, **$P < 0.01$, ***$P < 0.005$, ****$P < 0.001$. All error bars represent the standard deviation around the mean.

function demonstrated a reduction in the mitochondrial membrane-associated proteins VDAC1 and TOM20 that was dependent on the autophagy protein ATG5 consistent with the induction of mitophagy. Interestingly, devimistat induced mitophagy was not from a decline in MMP. The ability of devimistat to generate mitochondrial ROS is consistent with previous studies in NSCLC cells[8]. Separate studies have demonstrated that ROS can induce mitophagy independently of MMP[31] however, that

process involved the recruitment of Parkin. Devimistat-mediated mitophagy may be a ROS-dependent but Parkin-independent process not previously described. The ability of NAC to increase mitochondrial ROS has been previously reported[45] however the effects were seen shortly after NAC addition. How NAC is increasing mitochondrial ROS at 24 h in a devimistat-dependent manner is not clear. Additional studies will be needed to characterize this mechanism.

The age-related decline in mitochondrial quality leading to decreased MMP and increased sensitivity to devimistat combined with the age-related decline in autophagy make TCA cycle inhibition an ideal strategy to use in the treatment of older patients with cancer. Consistent with this are the results seen with devimistat in combination with chemotherapy in patients with metastatic pancreatic cancer[10]. Pancreatic cancer is also a disease of the elderly and the median age on that trial was 64. The in vivo results with metformin and devimistat results suggest a chemotherapy-free regimen could be translated to treat elderly patients. Such regimens have been tried with some success in B cell lymphomas[46] and could be adapted to AML patients. Indeed, studies of elderly populations have shown a correlation between decreased mitochondrial respiratory capacity in blood cells and physical frailty[47]. Thus, frail elderly patients who have the largest unmet medical need would be the most likely to benefit from this approach. Given the findings of decreased mitochondrial function in AML cells from older HSCs exploration of combined mitochondrial inhibition for the treatment of AML in the older patient is worthy of further exploration.

## Methods

**Study design**. Details of the phase I study were published previously[13]. The single-arm phase II study was approved by the Institutional Review Board (IRB) of Wake Forest Baptist Health. The complete protocol is provided in Supplementary Note 1. All patients who participated provided written informed consent on an IRB-approved consent form in accordance with the declaration of Helsinki. The study was conducted under the supervision of the Safety and Toxicity Review Committee of Wake Forest Baptist Health. The first patient was enrolled on October 27, 2015 and the last patient was enrolled on September 28, 2018.

**Eligibility**. Key inclusion criteria included histologically or cytological documented relapsed and/or refractory AML, age ≥18 years old, ECOG performance status of ≤3 and an expected survival of >3 months. The eligibility criteria were identical to the phase I study[13].

**Study treatment**. Devimistat at 2500, 2000, or 1500 mg/m$^2$ was administered via central line over 2 h on days 1–5. Cytarabine was given at 3 gm/m$^2$ for age <60 or 1.5 gm/m$^2$ for age ≥60 over 3 h every 12 h for five doses starting on day 3. Mitoxantrone was given at 6 mg/m$^2$ daily for three doses over 15 min after the 1st, 3rd, and 5th doses of cytarabine. Each patient was treated with one cycle of induction and could, at the discretion of the treating physician, receive a second course of induction or an abbreviated 3-day course. Any patient with significant residual disease after two cycles was considered refractory and removed from the study. Responding patients could receive up to two consolidation cycles of therapy. Any patient who completed all planned therapy could receive maintenance consisting of devimistat at 2500 mg/m$^2$ on days 1–5 of every 28 days until disease progression, withdrawal of consent, or the availability of allogenic stem cell transplant.

**Assessments**. Bone marrow biopsies were performed within 2 weeks of enrollment and on day 14 following each salvage induction cycle to assess for residual disease. Bone marrow biopsies were also performed to assess remission status when peripheral blood counts recovered to an absolute neutrophil count of >1000/μL and a platelet count of >100,000/μL or at the discretion of the treating physician. Responses were assessed as per ELN 2017 guidelines[2].

## Objectives

*Primary objective*. To determine the feasibility of devimistat when administered with high-dose cytarabine, and mitoxantrone in all three phases of salvage therapy (induction, consolidation, and maintenance). The regimen will be considered feasible if ≥50% of patients eligible for maintenance therapy complete at least three cycles.

*Secondary objectives*. To observe the response rate (CR and CRi) of CPI-613 in combination with high-dose cytarabine and mitoxantrone.

To observe the overall survival of patients treated with CPI-613 in combination with high-dose cytarabine and mitoxantrone in induction, consolidation, and maintenance.

To monitor toxicities experienced by patients treated with CPI-613 in combination with high-dose cytarabine and mitoxantrone in induction, consolidation, and maintenance.

**Mouse studies**. The Wake Forest University Institutional Animal Care and Use Committee approved all mouse experiments. For the syngeneic model, luciferase-tagged leukemia cells were transplanted into 6- to 8-wk-old sublethally irradiated (4 Gy) recipient C57Bl/6 mice (Jackson Laboratories) by tail-vein injection. Mice were monitored by bioluminescent imaging performed using an IVIS100 imaging system (Caliper LifeSciences, Hopkinton, MA). Mice were injected with 150 mg/kg D-Luciferin (Gold Biotechnology, St. Louis, MO), anesthetized with isoflurane, and imaged for 2 min. Devimistat with or without metformin was initiated upon detection of clear signals. For the PDX model, 8 week old sublethally irradiated (1 Gy) NSGS mice (Jackson Laboratories) were injected with 5 × 10$^6$ cells. Three days following injection mice were started on treatment with devimistat, metformin, or both. Mice were treated with devimistat by oral gavage. Metformin was added to the drinking water at 5 mg/100 ml. Animals were followed for survival. All mice utilized were female for ease of handling and to allow for housing of multiple animals per cage. Mice were kept up to five animals per cage. Mice were housed in a vivarium kept at 70° Fahrenheit and subjected to 12 h light/dark cycles.

**Cell culture**. RHRAS cells were kindly provided by Dr Gang Greg Wang, RN2 cells by Dr Christopher R. Vakoc. MFL2, RN2, and RHRAS cells were grown in stem cell media at 37 °C with 5% CO$_2$ as in ref. [13]. OCI-AML3 and K562 cells were maintained in RPMI media (Gibco) supplemented with 20% FBS, penicillin, and streptomycin. Cells were grown at 37 °C with 5% CO$_2$.

**TMRM staining/MitoSOX red staining**. Cell populations were exposed to the indicated therapy for 72 h and the viable cell population was determined using the Cell Titer Glo assay (Promega, Madison, WI) according to the manufacturer's protocol. For mitochondrial membrane potential assays cells were grown in stem cell media, placed at 200,000 cells/ml the night before and the next day cells were counted, and viability assessed by trypan blue exclusion. Cells were then stained with TMRM according to the manufacturer's protocol (ThermoFisher, Waltham, MA) and analyzed by flow cytometry as in ref. [13] using FCS Express software, version 7.12.0005. For mitochondrial ROS assessments, cells were treated as indicated and stained with MitoSOX Red according to the manufacturer's protocol (ThermoFisher, Waltham, MA).

**Cas9 Crisper gene deletion**. MFL2 and K562 cells were infected with the Cas9 expressing vector, MSCV_Cas9_puro (gift from Christopher Vakoc, Addgene plasmid # 65655) and infected cells selected with puromycin. Resistant cells were then transfected with sgRNA expressing vector LRG also a gift from Christopher Vakoc (Addgene plasmid # 65656) tagged with GFP targeting the indicated gene. Gene deletion was confirmed by western blot. Clonal populations of deleted cells were obtained by serial dilution. Primer sequences for sgRNAs are provided in Supplementary Table 3.

**Mitochondrial isolation and western blotting**. Cells were plated at a cell density of 300,000 viable cells per mL in 10-cm plates and treated with 100 μM CPI-613 (Devimistat) 24 h and 50 nM Bafilomycin A (Sigma #B1793) 2 h before harvest where indicated. Mitochondrial extracts were prepared with a Mitochondrial Isolation Kit for Cultured cells (Thermo Scientific #89874) as per the manufacturer's protocol. Pellets were lysed in Laemmli buffer, quantified by Bio-Rad Protein Assay (Bio-Rad), separated by SDS-PAGE, and transferred to an Immobilon PVDF membrane (Millipore). Antibodies against TOM20 (Cell Signaling, #42406; 1:1000), VDAC (Abcam, ab14734; 1:1000), and β-actin (Abcam, ab8227; 1:2000) were used.

**Confocal microscopy**. MFL2 cells were infected with pBMN-mCherry-Parkin, which was a gift from Richard Youle (Addgene plasmid #59419) and seeded on polylysine (Sigma Aldrich) covered four-well chamber slides. Then the cells were treated with 100 μM devimistat or 10 μM FCCP for 24 h. Cells were incubated with MitoTracker Deep Red (ThermoFisher, Waltham, MA) for 30 min at 37 °C. After washing with PBS, the cells were fixed with 10% formalin at room temperature for 20 min. After washing twice with PBS, cells were incubated with VECTASHIELD® mounting media with DAPI to stain the nucleus. Confocal microscopy was performed using an Olympus FV1200 SPECTRAL Laser scanning confocal microscope. All images were captured using the ×60 objective.

**Patient samples and Annexin V and PI staining**. All patient samples were collected during routine clinical care under a protocol approved by the Wake Forest School of Medicine Institutional Review Board. All patients provided written informed consent. For viability studies, mononuclear cells were isolated by Ficoll gradient separation and stored at −80 °C until use. Patient-derived AML cells were purified using CD34 magnetic beads (Miltenyi Biotec) and amplified by passage through NSGS mice (Jackson Laboratories). CD34+ cells were injected into sublethally irradiated (2 Gy) mice. Peripheral blood was monitored longitudinally for human CD33+ cells. Once substantial engraftment was recorded in the mice, the spleens and bone marrow were harvested and the mouse cells were depleted by staining with mouse CD45.1 and selected with APC beads (Miltenyi Biotec), and assessed the selection by flow cytometry. All the cells were divided into different treatments indicated in the figures and cell death was assessed by Annexin V

(BD Biosciences) and PI staining by flow cytometry at 72 h. Patient sample 19 was derived from a 33-year-old with poor risk AML at initial diagnosis. Patient sample 137 was derived from a 67-year-old at diagnosis with a normal karyotype AML. Patient sample 146 was derived from a 75-year-old at diagnosis with a secondary AML. Patient sample 210 was derived from a 56-year-old with therapy-related AML in the second relapse. Patient samples for Western blotting were collected during routine clinical care under a protocol approved by the Wake Forest School of Medicine Institutional Review Board. All patients provided written informed consent. The 33-year-old sample was from patient 19. The 53-year-old sample as from diagnosis with a normal karyotype AML. The 59-year sample was at diagnosis with a normal karyotype AML. The 89-year-old sample was at diagnosis with a poor risk karyotype AML.

**Quantitative PCR assays.** Cells were treated as indicated and total RNA harvested using RNeasy mini kits as per the manufacturer's protocol (Qiagen, Valencia, CA). RNA was converted to cDNA using the iScript cDNA synthesis kit (Bio-Rad, Hercules, CA). qPCR was carried out on a CFX-96 QPCR machine using SsoFast EvaGreen Mastermix as per the manufacturer's protocol (Bio-Rad). Relative message levels were calculated using the $\Lambda\Lambda$Ct method and normalized to actin. Primer sequences are available on request.

### RNA sequencing analysis

*Patient samples.* Samples from bone marrow biopsies at the time of enrollment were taken when possible. Mononuclear cells were isolated by Ficoll gradient separation and stored in liquid nitrogen. Response data were available for all patient samples. Total RNA was isolated from bone marrow-derived mononuclear cells using the Qiagen RNeasy RNA isolation kit per the manufacturer's instructions. RNA samples were assessed for quality by electrophoretic tracing (Agilent Bioanalyzer). cDNA libraries were generated using the Illumina TruSeq Stranded Total RNA kit with Ribo-Zero rRNA depletion according to the manufacturer's instructions. Library size distributions were inspected for quality using an Agilent 2100 Bioanalyzer. Library quantity was measured on a Qubit 3.0 (ThermoFisher, USA). Indexed libraries were pooled and sequenced to a target read depth of 40–50 M reads per library using 150 bp paired-end sequencing on an Illumina NextSeq 500 high-output flow cell. For each sample, >80% of sequences achieved >Q30 Phred quality scores (FASTQC analysis). Adapter contamination was cleaned with Trimmomatic[48] V0.32. Reads were aligned to the reference human genome GRCh38 (https://www.ncbi.nlm.nih.gov/assembly/GCF_000001405.39) using the STAR sequence aligner[49], and gene counts determined using featureCounts software[50] version 2.0.3. Differentially expressed genes were identified by negative binomial modeling using DESeq2[51] and false discovery correction ($q < 0.05$, Benjamini–Hochberg) Genes associated with patient age (by Pearson correlation) were identified by Gene Set Enrichment Analysis (GSEA)[18]. To confirm the association between patient age and mitochondrial gene expression in our bone marrow samples, the previously published OHSU AML RNAseq dataset of 239 baseline (pre-treatment) bone marrows[19] was analyzed. This dataset was accessed from cBioPortal (https://www.cbioportal.org/) in counts per million (CPM)-normalized format. This work was performed by the Cancer Genomics Shared Resource (CGSR) and the Bioinformatics Shared Resource (BISR) of the Wake Forest Baptist Comprehensive Cancer Center.

**Statistical analysis.** All means were compared by two-tailed ANOVA analysis with direct comparisons done with a Sidak's multiple comparison test. A Sidak's multiple comparisons test adjusts the $P$ value to account for three or more comparisons. When only two means were compared a two-tailed Student's $t$ test was utilized. Survival curves were estimated by the Kaplan–Meier method and $P$ values were determined by the log-rank test. Adjusted $P$ values below 0.05 were considered significant. Demographics were compared by Exact tests, Kruskal–Wallis tests, and log-rank test to compare survival among the dose groups of devimistat, both in the 60+ group and all participants. Patients were followed and analyzed for response rate and overall survival. Analysis was performed using Graph Pad Prism version 8.3.0 (Graph Pad Software Inc). Combinatorial indices were calculated using Calcusyn version 2.0 (Biosoft, Cambridge, UK).

**Reporting summary.** Further information on research design is available in the Nature Research Reporting Summary linked to this article.

## Data availability

The reference human genome GRCh38 can be found at https://www.ncbi.nlm.nih.gov/assembly/GCF_000001405.39. Raw and processed RNAseq data generated in this study can be accessed through the Gene Expression Omnibus (GSE195933). Additional de-identified patient data can be requested from the corresponding author until January 30, 2025. Source data are provided with this paper.

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

## Acknowledgements

Support was provided by the Genomics Shared Resource facility Director Lance D. Miller, PhD, and facility personnel. We also thank Sanjeev Luther and Howard Jonas from Rafael Pharmaceuticals for insightful comments and CPI-613 for both preclinical and clinical work. We thank Navdeep S. Chandel and Enzo Palma for critical reading of the manuscript. TSP, RGA, and KMP are supported by the National Cancer Institute award 1R01CA197991-01A1. The Wake Forest Baptist Comprehensive Cancer Center Cancer Genomics Shared Resource, and the Biostatistics and Bioinformatics Shared Resources contributed to this work and are supported by the National Cancer Institute's Cancer Center Support Grant award number P30CA012197. The content is solely the responsibility of the authors and does not necessarily represent the official views of the National Cancer Institute. The funders had no role in study design, data collection, and analysis or manuscript writing.

## Author contributions

R.A. designed research, performed research, analyzed the data, and edited the manuscript; L.D.M. analyzed the data and edited the manuscript; S.I. analyzed the data; J.W.C. performed research, analyzed the data, and edited the manuscript; K.P.M. performed research and analyzed the data; N.J.S. performed research and analyzed the data; L.R.E. analyzed the data and edited the manuscript; D.S.H. analyzed the data and edited the manuscript; R.R.B. analyzed the data and edited the manuscript; M.M. analyzed the data and edited the manuscript; S.D. analyzed the data and edited the manuscript; S.L. analyzed the data and edited the manuscript; B.L.P. analyzed data and edited the manuscript; T.S.P. designed research, analyzed the data, and wrote the manuscript.

## Competing interests

T.S.P. and B.L.P. are paid consultants of Rafael Pharmaceuticals and T.S.P. is Co-Chief Medical Officer. T.S.P. receives research funding from Rafael Pharmaceuticals who owns the licensing rights to CPI-613 and is currently developing it for use in oncology patients. Rafael Pharmaceuticals had no input over the design of the protocol as it was initiated as an investigator-sponsored trial by T.S.P. prior to his consultancy and role as Co-Chief Medical Officer. Rafael Pharmaceuticals had no approval rights over the manuscript. The remaining authors declare no competing interests.
