## [Peer Review File · Nature Communications]

Phase II trial of cytarabine and mitoxantrone with devimistat
in acute myeloid leukemiaREVIEWER COMMENTS

Reviewer #1 (Remarks to the Author); expert on preclinical and clinical AML and mitochondria:

Anderson et al., have evaluated the lipoate analogue, devimistat, in AML. They have included data from phase I/II trials as well as preclinical studies to determine the mechanism of action of the drug. The paper has several sections that range from predictors of clinical response to mechanism of action. While the data are interesting, I think further experimental evidence would be needed to support many of their conclusions.

Specific comments include:

- 1) The authors performed RNA sequencing on primary AML samples from their phase I/II trial of devimistat. They isolated RNA from total marrow samples rather than from isolated AML blasts. From these gene expression studies, they conclude that AML blasts from older patients have decreased mitochondrial function compared to younger patients. Although an interesting conclusion, I think more evidence is required to make this claim. The percentage of blasts in these patients was less than 50% and variable between patients. Given the lower blast percentage, residual normal hematopoietic and stromal cells will most likely be a significant contributor to their gene analysis. As such, it is unclear whether differences in gene expression based on age relate more to differences in AML blasts or differences that arise in normal cells as individuals age. To address this point, the authors can compare larger gene expression data sets from AML patients and relate the expression to age.
- 2) The authors can test the impact of aging on mitochondrial function using gene expression in mouse marrow from young and old mice.
- 3) Related to the above points, the authors conclude that there is a significant decline in mitochondrial function of leukemic cells with age based on changes in gene expression. To support this conclusion, I think they should conduct functional studies with primary AML cells and relate mitochondrial function to age. They can examine basal OCR, reserve capacity, complex activity, MMP, etc. Complementary studies with normal bone marrow cells from young and old individuals should also be performed.
- 4) The authors claim that older patients derive most benefit from devimistat as their leukemic cells show the least expression of mitochondrial genes. But, one could argue that cells with the lowest expression of mitochondrial genes would be the least reliant on oxphos and the least sensitive to devimistat. So, the gene expression would not automatically predict function or sensitivity. In order to make this conclusion, I think the authors should directly test the sensitivity to the drug in primary AML blasts isolated from young and old patients.
- 5) Related to the above section, they also claim that MMP changes with age. I think they should also directly measure MMP in patient samples and relate to age.
- 6) The authors demonstrate synergy with metformin + devimistat and propose mechanisms related to changes in MMP. However, devimistat also increased ROS production. Is it possible, the combination of devimistat and metformin are leading to increased ROS production and that is the primary driver of cell death, rather than changes in MMP? I think the authors should measure ROS with the combination and determine whether the cell death is prevented by NAC.
- 7) Related to the above point, they should test devimistat in combination with other ROS inducing agents and other agents that increase or decrease MMP.
- 8) Is the increased ROS produced by devimistat driving the increased autophagy?
- 9) The authors treat cells with doxycycline to inhibit mitochondrial protein synthesis and show increased sensitivity to devimistat. However, doxycycline may be acting by inhibiting oxidative phosphorylation rather than changing mitochondrial turnover. Additional functional experiments are needed to demonstrate that doxycycline is acting by changing mitochondrial turnover rather than on oxphos.
- 10) The in vivo effects of the metformin and devimistat are modest. The authors should test the combination in mice engrafted with primary AML cells.

Reviewer #2 (Remarks to the Author); expert on translational research and clinical trials for AML:

The manuscript “Devimistat Leverages Age Related Vulnerabilities in Acute Myeloid Leukemia Cells by Inducing Mitochondrial Turnover and Chemosensitization Benefiting Older Patients” by Anderson, et.al., is a continued investigation of treatment with devimistat in AML. Devimistat is a lipoate derivative that inhibits pyruvate dehydrogenase and α -ketoglutarate dehydrogenase. Combining data from a phase I and phase II trial, the authors demonstrate a dose response in older but not younger patients. Baseline bone marrow MNCs from a small subset of patients (N=25) were analyzed by RNAseq. GSEA showed an age-related decline of mitochondrial function (cellular respiration, oxphos, electron transport chain, and others). In additional studies, mostly in AML cell lines, they show that high MMP confers resistance to devimistat. In sensitive cell lines they show that inhibition with devimistat results in increased reactive oxygen species and mitochondrial turnover in AML cell lines. Consistent with this, devimistat effects were enhanced by inhibition of autophagy (chloroquine or deletion of Atg5). They conclude that age-related mitochondrial decline leads sensitizes leukemia cells from older AML patients to devimistat, while no such effect is apparent in younger patients.

Major Concerns:

1. The analysis of the clinical trial is post hoc, not planned and as such only suggestive. Validation in an independent cohort is required. This critical limitation is not discussed.
2. The overall design of the clinical trial is difficult to appreciate from the text and even the comfort diagram is not fully informative. Please provide schematic that clearly shows when which drug was given, including chemotherapy. Ditto it is difficult to appreciate which prior therapies patients had been treated with.
3. How were samples selected for RNAseq? Please provide information about these patients. Somatic mutations?
4. Some key data are too limited to support the conclusions. For instance, while the correlation between MMP and devimistat sensitivity looks suggestive, it is based on only three cell lines that are genetically very different. Additional cell lines should be tested to validate the suspected association.
5. Please demonstrate that metformin has no synergistic effects in MMP low cell lines.
6. Little to no data provided on primary patient samples. One would like to see additional data regarding mitochondrial membrane potential and mitochondrial turnover in primary AML patient samples.
7. Non non-leukemic controls were included. One would like to see further data in regards to impact on healthy hematopoietic cells.
8. There are no salvage experiments. For instance can they rescue the cells by overexpression of PDH and AKGDH?
9. Doxycycline inhibits mitochondrial translation at the 10 microM concentration used in these experiments (Moullan et al. Cell Rep. 2015;10(10):1681-1691. PMID: 25772356). The notion of dox as a mitochondrial biogenesis inhibitor seems overly simplistic.

Minor Concerns:

1. Unlike other figures, Figure 6 does not have statistical summary included in figure legend.
2. Figure 6 D-E graphs axis and head titles should be more descriptive.
3. It would be useful to see additional data confirming changes in mitochondrial function such as changes in ATP concentration upon treatment with devimistat.

Reviewer #3 (Remarks to the Author); expert on biostatistics and clinical trial design:

The authors write a well described and detailed study to assess the mechanism of action of devimistat, a drug which is postulated to be used for treatment of AML in older patients. An analysis using the combination of phase I and phase II trial data showed that dose response only existed in older and not in younger population. This could be caused by TCA cycle inhibition causing ROS which in turn cause mitophagy. Older patients may have decrease autophagy capacity and functional mitochondrial decline which makes them sensitive to the effects of devimistat. Older patients with AML typically have poorer survival outcomes when compared to the younger population and hence could benefit from devimistat.

Here are few concerns:

Under eligibility criteria, one of them is the histologically and cytologically established relapsed AML. Does this mean this is a next step intervention among patients of AML who already tried other drugs? If yes, it may be helpful to list what was tried before.

The primary objectives and secondary objectives are stated in a manner which explains the purpose for which the original trial was conducted. However, in this study, the primary objective is actually to explore the dose response and study the mechanism of action of devimistat which is conducive to older patients. This is also the subject of the entire discussion. There should be clarity when stating objectives and must be held constant across the paper, in order to avoid confusion to the reader. Some restructuring and re-alignment of the manuscript will be helpful.

Before the use of an abbreviation, the full form must be stated. For example RHRAS to mention one. However, there are several instances where this occurs.

In the statistical analysis section, it is stated that a p value of <0.05 will be considered as statistically significant. However, the fact that multiple comparison adjustment was considered is not explicitly stated. Furthermore why was multiple comparison adjustment not considered in certain instances such as for figures 3, 4 and 6 – more so for the latter two because multiple comparison adjustment would have invalidated the findings with single asterisk (which means p is only less than 0.05)

A brief explanation of what is meant by combinatorial index will be helpful.

Results:

The first paragraph should be clearly delineated as baseline characteristics, if that is what is intended to be shown.

In supplemental table 2, what are the percentages based off of? For example: $18+10+1+7=36$ which is not 100% of sample size of 48.

In table 1: adding a p value to comparison of characteristics using appropriate tests is essential and should be stated in the stat analysis section. Columns denoting 'all' and '>= 60 years old' can be merged.

Line 304: What was the number of patients greater than 60 years?

Matching based on demographic factors must be a strength. The use of the word 'Despite' in this instance is incorrect. You could just say: Despite the small numbers, there was a significant advantage...

However, here again, multiple comparison adjustment must be done. Because you are using several Kaplan Meier Curves.

In figures 2 A-E, all graphs show a decrease in enrichment scores (green line) as they pass along the X-axis (which is constituted by positively correlated genes on left negatively correlated on right), after which there is a steep increase. I am confused as to how these graphs show an age related decline in expression of these genes. A clarification will be very helpful in this regard.

Line 347: Combinatorial index 0.67 considered as fair, moderate, good...?

In the section on Autophagy and Devimistat: If Devimistat induces mitochondrial fission and MDIVI-1 inhibits it, how is the combination sensitizing AML cells to the effects of Demovistat?

Similar question with Chloroquine and Devimistat w.r.t autophagy.

Also, an explanation of why the constancy of TOM20 and VDAC protein (when Atg5 deleted cells were treated with devimistat) is important will be helpful..

Reviewer #1 (Remarks to the Author); expert on preclinical and clinical AML and mitochondria:

Anderson et al., have evaluated the lipoate analogue, devimistat, in AML. They have included data from phase I/II trials as well as preclinical studies to determine the mechanism of action of the drug. The paper has several sections that range from predictors of clinical response to mechanism of action. While the data are interesting, I think further experimental evidence would be needed to support many of their conclusions.

We thank the reviewer for their frank assessment.

Specific comments include:

1) The authors performed RNA sequencing on primary AML samples from their phase I/II trial of devimistat. They isolated RNA from total marrow samples rather than from isolated AML blasts. From these gene expression studies, they conclude that AML blasts from older patients have decreased mitochondrial function compared to younger patients. Although an interesting conclusion, I think more evidence is required to make this claim. The percentage of blasts in these patients was less than 50% and variable between patients. Given the lower blast percentage, residual normal hematopoietic and stromal cells will most likely be a significant contributor to their gene analysis. As such, it is unclear whether differences in gene expression based on age relate more to differences in AML blasts or differences that arise in normal cells as individuals age. To address this point, the authors can compare larger gene expression data sets from AML patients and relate the expression to age.

We thank the reviewer for this insightful comment. We agree and have completed an analysis of the previously published OHSU AML RNAseq data set of 239 baseline (pre-treatment) bone marrows (PMID: 30333627). We found the same age related decline in mitochondrial genesets and revised the manuscript (new figure 2).

2) The authors can test the impact of aging on mitochondrial function using gene expression in mouse marrow from young and old mice.

We agree and thank the reviewer for their excellent suggestion. We have generated AML models from old and young murine hematopoietic stem cells (HSCs) and shown a decrease in mitochondrial gene expression, basal OCR, mitochondrial content and MMP in AML cells derived from older HSCs consistent with our observations in the human cohorts (see new supplemental figure 2).

3) Related to the above points, the authors conclude that there is a significant decline in mitochondrial function of leukemic cells with age based on changes in gene expression. To support this conclusion, I think they should conduct functional studies with primary AML cells and relate mitochondrial function to age. They can examine basal OCR, reserve capacity, complex activity, MMP, etc. Complementary studies with normal bone marrow cells from young and old individuals should also be performed.

We agree and have attempted to address this with the experiments described above using murine models. We also agree that primary patient sample data would further strengthen this point unfortunately in our experience primary patient samples must be freshly harvested from host mice in order to be reliably metabolically active. Cells thawed and placed in culture are not robustly metabolically active in our experience and do not provide reproducible data. As we do not currently have the resources to allow us to conduct a robust study of primary patient samples we have adjusted the language in the manuscript to moderate the conclusion.

4) The authors claim that older patients derive most benefit from devimistat as their leukemic cells show the least expression of mitochondrial genes. But, one could argue that cells with the lowest expression of mitochondrial genes would be the least reliant on ox/phos and the least sensitive to devimistat. So, the gene expression would not automatically predict function or sensitivity. In order to make this conclusion, I think the authors should directly test the sensitivity to the drug in primary AML blasts isolated from young and old patients.

We thank the reviewer for their comment and apologize for any lack of clarity. It is certainly true that there are differences in the degree of utilization of and dependency on ox/phos for AML cells at baseline for the purposes of cell replication however, when stressed with chemotherapy or anti-BCL2 agents AML cells utilize mitochondrial metabolism as a source of resistance (see multiple papers including PMID 31666400 and PMID 28416471 and reviewed in PMID 29117545). We were therefore suggesting that mitochondrial capacity likely plays a role resistance to combination regimens that include a mitochondrial targeted agent. Again, we agree that direct patient sample experiments would strengthen the manuscript and regret our inability to conduct a robust analysis of primary patient samples. We have moderated the language in the paper.

5) Related to the above section, they also claim that MMP changes with age. I think they should also directly measure MMP in patient samples and relate to age.

Again, we agree with the review that such data would strengthen the manuscript but are limited in our ability to conduct these experiments for the reasons listed under point 3. We did conduct this experiment in murine models as outlined in point 2 which directly address MMP in AML from younger and older HSCs.

6) The authors demonstrate synergy with metformin + devimistat and propose mechanisms related to changes in MMP. However, devimistat also increased ROS production. Is it possible, the combination of devimistat and metformin are leading to increased ROS production and that is the primary driver of cell death, rather than changes in MMP? I think the authors should measure ROS with the combination and determine whether the cell death is prevented by NAC.

We agree and thank the reviewer for their comment. We have completed the suggested study and see that there is an increase in mitochondrial ROS when metformin and devimistat are administered together. To assess the functional consequence of this we assessed the ability of

NAC to rescue as you have suggested. NAC was unable to rescue cells (see new supplemental figure 3). Additionally, further experiments with the complex III inhibitor antimycin A also displayed sensitization suggesting a direct effect of ETC inhibition (see new supplemental figure 3).

7) Related to the above point, they should test devimistat in combination with other ROS inducing agents and other agents that increase or decrease MMP.

We agree and thank the reviewer for their comment. We have completed the suggested studies. Devimistat activity was increased with the complex III inhibitor antimycin A as well as the ROS generating paraquat (see new supplemental figure 3). These data indicate that sensitivity to devimistat can be induced by other ETC inhibitors as well as by increasing ROS.

8) Is the increased ROS produced by devimistat driving the increased autophagy?

Yes, we believe the mitochondrial ROS produced by devimistat is the trigger for mitophagy and show that NAC treatment stabilizes the levels of TOM20 and VDAC (figure 6E).

9) The authors treat cells with doxycycline to inhibit mitochondrial protein synthesis and show increased sensitivity to devimistat. However, doxycycline may be acting by inhibiting oxidative phosphorylation rather than changing mitochondrial turnover. Additional functional experiments are needed to demonstrate that doxycycline is acting by changing mitochondrial turnover rather than on oxphos.

We agree and have conducted additional experiments showing that doxycycline does not significantly effect the mitochondrial membrane potential (new supplemental figure 4).

10) The in vivo effects of the metformin and devimistat are modest. The authors should test the combination in mice engrafted with primary AML cells.

We agree and were able to conduct a small PDX model experiment. This data has now been included in supplemental figure 3 and in the results section.

Reviewer #2 (Remarks to the Author); expert on translational research and clinical trials for AML:

The manuscript "Devimistat Leverages Age Related Vulnerabilities in Acute Myeloid Leukemia Cells by Inducing Mitochondrial Turnover and Chemosensitization Benefiting Older Patients" by Anderson, et.al., is a continued investigation of treatment with devimistat in AML. Devimistat is a lipoate derivative that inhibits pyruvate dehydrogenase and α -ketoglutarate dehydrogenase. Combining data data from a phase I and phase II trial, the authors demonstrate a dose response in older but not younger patients. Baseline bone marrow MNCs from a small subset of patients (N=25) were analyzed by RNAseq. GSEA showed an age-related decline of

mitochondrial function (cellular respiration, oxphos, electron transport chain, and others). In additional studies, mostly in AML cell lines, they show that high MMP confers resistance to devimistat. In sensitive cell lines they show that inhibition with devimistat results in increased reactive oxygen species and mitochondrial turnover in AML cell lines.

Consistent with this, devimistat effects were enhanced by inhibition of autophagy (chloroquine or deletion of Atg5). They conclude that age-related mitochondrial decline leads sensitizes leukemia cells from older AML patients to devimistat, while no such effect is apparent in younger patients.

We thank the reviewer for their assessment.

Major Concerns:

1. The analysis of the clinical trial is post hoc, not planned and as such only suggestive. Validation in an independent cohort is required. This critical limitation is not discussed.

We agree and apologize for not stating this limitation more directly. We have amended the discussion section (line 559).

2. The overall design of the clinical trial is difficult to appreciate from the text and even the comfort diagram is not fully informative. Please provide schematic that clearly shows when which drug was given, including chemotherapy. Ditto it is difficult to appreciate which prior therapies patients had been treated with.

We thank the reviewer for their comment. We have now included a figure with the treatment schema for the trial (supplemental figure 1). As for prior therapy in the phase II study, in the frontline setting 43/48 (90%) patients initial therapy was with 7+3, 2 patients were treated initially with vyxeos, 2 with clofarabine and 1 with hypomethylating agent. This has been added to the results section.

3. How were samples selected for RNAseq? Please provide information about these patients. Somatic mutations?

All samples with sufficient quality RNA were included in the RNA sequencing analysis.

4. Some key data are too limited to support the conclusions. For instance, while the correlation between MMP and devimistat sensitivity looks suggestive, it is based on only three cell lines that that are genetically very different. Additional cell lines should be tested to validate the suspected association.

We thank the review for their observations. We have now conducted additional experiments to support our hypothesis regarding the relationship between MMP and response to deveimistat (please see new figure 3, supplemental figure 3).

5. Please demonstrate that metformin has no synergistic effects in MMP low cell lines.

We thank the review for the excellent suggestion. This experiment has been incorporated into new supplemental figure 3C.

6. Little to no data provided on primary patient samples. One would like to see additional data regarding mitochondrial membrane potential and mitochondrial turnover in primary AML patient samples.

We agree and thank the reviewer for their comment. We have included additional experiments utilizing primary patient samples including assessments of ETC protein expression, mitochondrial ROS generation and an in vivo PDX experiment (see figures 5H, 6D, supplemental figures 2A+B, 3F, and 5C).

7. Non non-leukemic controls were included. One would like to see further data in regards to impact on healthy hematopoietic cells.

We agree with the reviewer that toxicity of devimistat on normal hematopoietic cells is an important consideration. We would point the reviewer to the safety data (results section lines 282-298) that show no increase in hematologic toxicities over what would be expected from a high dose cytarabine and mitoxantrone-containing regimen suggesting an acceptable safety profile.

8. There are no salvage experiments. For instance can they rescue the cells by overexpression of PDH and AKGDH?

We agree with the reviewer that rescue by overexpression of PDH or KGDH would strengthen the manuscript unfortunately they are both multi-protein complexes that would require the simultaneous expression of the respective E1, E2 and E3 subunits and is therefore not feasible.

9. Doxycycline inhibits mitochondrial translation at the 10 microM concentration used in these experiments (Moullan et al. Cell Rep. 2015;10(10):1681-1691. PMID: 25772356). The notion of dox as a mitochondrial biogenesis inhibitor seems overly simplistic.

We agree and thank the reviewer for their comment. We have modified the text describing the effect of doxycycline as an inhibitor of mitochondrial protein translation and incorporated the references provided.

Minor Concerns:

1. Unlike other figures, Figure 6 does not have statistical summary included in figure legend.

We apologize for the oversight and have now added a statistical summary to the figure legend.

2. Figure 6 D-E graphs axis and head titles should be more descriptive.

Changed as requested.

3. It would be useful to see additional data confirming changes in mitochondrial function such as changes in ATP concentration upon treatment with devimistat.

We thank the reviewer for the excellent suggestion and have completed the requested study showing that devimistat directly reduces the amount of ATP in AML cells in a dose dependent fashion (see new figure 3A).

Reviewer #3 Comments

The authors write a well described and detailed study to assess the mechanism of action of devimistat, a drug which is postulated to be used for treatment of AML in older patients. An analysis using the combination of phase I and phase II trial data showed that dose response only existed in older and not in younger population. This could be caused by TCA cycle inhibition causing ROS which in turn cause mitophagy. Older patients may have decrease autophagy capacity and functional mitochondrial decline which makes them sensitive to the effects of devimistat. Older patients with AML typically have poorer survival outcomes when compared to the younger population and hence could benefit from devimistat.

Here are few concerns:

Under eligibility criteria, one of them is the histologically and cytologically established relapsed AML. Does this mean this is a next step intervention among patients of AML who already tried other drugs? If yes, it may be helpful to list what was tried before.

We thank the reviewer for their excellent suggestion. We have added detail to the previous regimens in the results section.

The primary objectives and secondary objectives are stated in a manner which explains the purpose for which the original trial was conducted. However, in this study, the primary objective is actually to explore the dose response and study the mechanism of action of devimistat which is conducive to older patients. This is also the subject of the entire discussion. There should be clarity when stating objectives and must be held constant across the paper, in order to avoid confusion to the reader. Some restructuring and re-alignment of the manuscript will be helpful.

We apologize for any confusion. We have separated the clinical trial results into the first portion of the results and added additional language to assist the reader.

Before the use of an abbreviation, the full form must be stated. For example RHRAS to mention one. However, there are several instances where this occurs.

We thank the reviewer for their comment and have defined each abbreviation at its first use in the manuscript. We apologize for any confusion as RHAS, MFL2, OCI and RN2 are not abbreviations but rather names of cell lines. We have clarified this in the text.

In the statistical analysis section, it is stated that a p value of <0.05 will be considered as statistically significant. However, the fact that multiple comparison adjustment was considered is not explicitly stated. Furthermore why was multiple comparison adjustment not considered in certain instances such as for figures 3, 4 and 6 – more so for the latter two because multiple comparison adjustment would have invalidated the findings with single asterisk (which means p is only less than 0.05)

We thank the reviewer for this comment and apologize for the confusion. All multiple comparisons were done with a two tailed ANOVA followed by a Sidak's adjustment for false discovery of multiple comparisons. This is now in the figure legends and the methods section.

A brief explanation of what is meant by combinatorial index will be helpful.

A combinatorial index is a mathematical model that allows for the quantification of the degree of synergy. This is now added to the text.

Results:

The first paragraph should be clearly delineated as baseline characteristics, if that is what is intended to be shown.

This has been added to the title of the section.

In supplemental table 2, what are the percentages based off of? For example: $18+10+1+7=36$ which is not 100% of sample size of 48.

We apologize for the confusion. All toxicities are reported by grade and patients may have experienced related toxicities captured under the same heading. For this reason, many headings have more instances than patients treated. All percentage of effected patients calculations were done by the number treated at that dose. This has now been added to the results section.

In table 1: adding a p value to comparison of characteristics using appropriate tests is essential and should be stated in the stat analysis section.

We thank the reviewer for this comment and have now added p values to table 1. This has also been added to the stats section as requested.

Columns denoting 'all' and '>= 60 years old' can be merged.

We thank the reviewer but respectfully do not agree. The cohorts described under each heading are not identical as the cohorts >=60 are a subset of the total cohort (listed under all). We felt the reader would want to see the characteristics of the entire population as well as those >=60.

Line 304: What was the number of patients greater than 60 years?

23 treated with 2,000mg/m² and 19 treated with 1,500mg/m². This has been added to the text.

Matching based on demographic factors must be a strength. The use of the word 'Despite' in this instance is incorrect. You could just say: Despite the small numbers, there was a significant advantage... However, here again, multiple comparison adjustment must be done. Because you are using several Kaplan Meier Curves.

We thank the review for their critique. In all panels in figure 1 there are 2 Kaplan-Meier curves being compared by log rank test. We have changed the language as suggested. Additionally, we have now stressed the post-hoc nature of the analysis and indicated that it is hypothesis generating only and that randomized data is needed.

In figures 2 A-E, all graphs show a decrease in enrichment scores (green line) as they pass along the X axis (which is constituted by positively correlated genes on left negatively correlated on right), after which there is a steep increase. I am confused as to how these graphs show an age related decline in expression of these genes. A clarification will be very helpful in this regard.

We thank the reviewer for this observation. In Gene Set Enrichment Analysis (<https://www.gsea-msigdb.org/gsea/index.jsp>), the X axis represents all genes ranked by a unit of measure. In figure 2 the genes are ranked by their expression pattern's correlation (R) with patient age (in the patient cohort). The steep increase on right identifies genes (predefined by biological category) with increasingly negative expression correlation with age. Please see PMID 16199517 for a detailed description of the method. We have added a general discretion of the method to the results section.

Line 347: Combinatorial index 0.67 considered as fair, moderate, good...?

CI values below 0.7 are considered synergistic. This has been added to the text.

In the section on Autophagy and Devimistat: If Devimistat induces mitochondrial fission and MDIVI-1 inhibits it, how is the combination sensitizing AML cells to the effects of Demovistat?

We apologize for the lack of clarity. The cellular response to devimistat is to induce mitophagy which in turn requires mitochondrial fission. It is a source of resistance to the effects of devimistat and when mitochondrial fission is inhibited by MDIVI-1 the cell becomes more sensitive.

Similar question with Chloroquine and Devimistat w.r.t autophagy.

Again, we apologize for the lack of clarity. As noted above the cellular response to devimistat is induction of mitophagy which requires BOTH mitochondrial fission and autophagy. Chloroquine inhibits late autophagy and therefore prevents efficient mitophagy leading to increased sensitivity.

Also, an explanation of why the constancy of TOM20 and VDAC protein (when Atg5 deleted cells were treated with devimistat) is important will be helpful..

The finding of preserved TOM20 and VDAC protein levels in cells that have a genetic impairment to autophagy is an orthotopic demonstration of the importance of autophagy in the devimistat induced mitochondrial turnover and further supports that devimistat induces mitophagy.

We thank all three reviewers for their insights, helpful suggestions and critiques.

REVIEWER COMMENTS

Reviewer #1 (Remarks to the Author):

Anderson et al have submitted a revised manuscript investigating devimistat in AML. The authors have conducted additional experiments to address the original concerns. However, in many places the new experiments seem to lack a sufficient sample size or important controls.

Specifically:

- 1) The immunoblot studies showing changes in ETC subunits with age is very interesting and an important figure to strengthen their conclusions of decline in ox-phos activity in AML with age. However, only 4 samples of variable ages were tested. I think a larger sample size needs to be tested to draw this conclusion. If the higher ETC expression in the 33 year old patient is not reproducible, the differences between the rest of the samples is very modest.
- 2) The murine study examining AML development after transducing HSC from young and old mice was an excellent experiment and I agree with the authors that this experiment can replace the need for functional studies on samples from young and old AML patients. There results are very striking and the difference is large. However, it seems that only one AML clone was isolated from the young mouse and one clone was isolated from the old mouse. I think it would be helpful to ensure the experiment is reproducible with more than 1 clone and the differences are related to age and not variability between clones
- 3) The authors show that NAC does not block cell death after the combination of devimistat and ROS inducing agents and thus conclude that the increase in ROS is not functionally important. I think an important control is to show that NAC decreases ROS production after these treatments. Without that control, it is hard to interpret the effects of NAC on cell viability.
- 4) The authors treated a primary patient samples with the combination of metformin and devimistat and claim there was a benefit to the combination. While there was a very slight increase in survival, there was also a higher early death rate. I don't think the data support their conclusion of in vivo benefit to the drug combination. Thus, the overall in vivo benefits to this combination are extremely modest, at best.

Reviewer #2 (Remarks to the Author):

To whom it may concern:

The authors have made substantial efforts to improve the paper, including additional experimental work. The use of many chemical inhibitors rather than more rigorous genetic approaches remains a general weakness, but altogether I think the observation of decreased mitochondrial function in older AML patients as a driver of improved response to TCA disruption is interesting enough to warrant publication.

Thank you for allowing me to review this paper.

Mike Deininger

Reviewer #3 (Remarks to the Author):

Minor comments:

Statistical analysis section:

I am not aware of what Sidak's multiple comparison test is. Perhaps a very brief mention of what it does will be helpful.

Say adjusted p values below 0.05 were considered significant

Say log-rank test to compare survival among the dose groups....

General text figures: in the figure where it says p-value is calculated by log rank test, please rephrase it to read something such as 'log rank test was used to assess difference between survival curves...' because the way the sentence is framed seems a little off. You can mention later 'An p value of <0.05 was considered significant'.

Supplemental table 1:

Indicate what is being described for line of salvage and performance score i.e. mean, median etc.

With regard to your last response to my query:

'The finding of preserved TOM20 and VDAC protein levels in cells that have a genetic impairment to autophagy is an orthotopic demonstration of the importance of autophagy in the devimistat induced mitochondrial turnover and further supports that devimistat induces mitophagy.'

Could you state the above in the manuscript so readers can understand.

Reviewer #1 (Remarks to the Author):

Anderson et al have submitted a revised manuscript investigating devimistat in AML. The authors have conducted additional experiments to address the original concerns. However, in many places the new experiments seem to lack a sufficient sample size or important controls.

We thank the reviewer for their frank assessment.

Specifically:

1) The immunoblot studies showing changes in ETC subunits with age is very interesting and an important figure to strengthen their conclusions of decline in ox-phos activity in AML with age. However, only 4 samples of variable ages were tested. I think a larger sample size needs to be tested to draw this conclusion. If the higher ETC expression in the 33 year old patient is not reproducible, the differences between the rest of the samples is very modest.

We thank the reviewer for their suggestion and have now completed additional Western blots and quantitated data from another 6 patient samples. We have quantitated data for 6 patients younger than 50 and 4 samples from patients older than 50 and quantitated the relative expression. The new data has been added to supplemental figure 2.

2) The murine study examining AML development after transducing HSC from young and old mice was an excellent experiment and I agree with the authors that this experiment can replace the need for functional studies on samples from young and old AML patients. There results are very striking and the difference is large. However, it seems that only one AML clone was isolated from the young mouse and one clone was isolated from the old mouse. I think it would be helpful to ensure the experiment is reproducible with more than 1 clone and the differences are related to age and not variability between clones

We agree and have now generated additional leukemias from adolescent and middle aged murine HSCs. Consistent with the previous data adolescent derived leukemia cells had a higher mitochondrial content and MMP demonstrating the same phenomenon in biologically independent replicates.

3) The authors show that NAC does not block cell death after the combination of devimistat and ROS inducing agents and thus conclude that the increase in ROS is not functionally important. I think an important control is to show that NAC decreases ROS production after these treatments. Without that control, it is hard to interpret the effects of NAC on cell viability.

We thank the reviewer for their excellent suggestion and have now completed the requested studies. Interestingly NAC did not attenuate mitochondrial ROS and this result has now been incorporated into the revised manuscript.

4) The authors treated a primary patient samples with the combination of metformin and devimistat and claim there was a benefit to the combination. While there was a very slight increase in survival, there was also a higher early death rate. I don't think the data support their conclusion of in vivo benefit to the drug combination. Thus, the overall in vivo benefits to this combination are extremely modest, at best.

We thank the reviewer for their frank assessment. We have stated the benefit is modest in the text.

Reviewer #2 (Remarks to the Author):

To whom it may concern:

The authors have made substantial efforts to improve the paper, including additional experimental work. The use of many chemical inhibitors rather than more rigorous genetic approaches remains a general weakness, but altogether I think the observation of decreased mitochondrial function in older AML patients as a driver of improved response to TCA disruption is interesting enough to warrant publication.

Thank you for allowing me to review this paper.

Mike Deininger

We thank Dr Deininger for his time in reviewing our paper.

Reviewer #3 (Remarks to the Author):

Minor comments:

Statistical analysis section:

I am not aware of what Sidak's multiple comparison test is. Perhaps a very brief mention of what it does will be helpful.

A Sidak's multiple comparisons test adjusts the P value to account for 3 or more comparisons. This has been added to the stats section.

Say adjusted p values below 0.05 were considered significant

Changed as requested.

Say log-rank test to compare survival among the dose groups....

Changed as requested.

General text figures: in the figure where it says p-value is calculated by log rank test, please rephrase it to read something such as 'log rank test was used to assess difference between survival curves...' because the way the sentence is framed seems a little off. You can mention later 'An p value of <0.05 was considered significant'.

Changed as requested.

Supplemental table 1:

Indicate what is being described for line of salvage and performance score i.e. mean, median etc.

With regard to your last response to my query:

'The finding of preserved TOM20 and VDAC protein levels in cells that have a genetic impairment to autophagy is an orthotopic demonstration of the importance of autophagy in the devimistat induced mitochondrial turnover and further supports that devimistat induces mitophagy.'

Could you state the above in the manuscript so readers can understand.

This has been added to the results section.

REVIEWER COMMENTS

Reviewer #1 (Remarks to the Author):

All my original concerns have been addressed

Reviewer #3 (Remarks to the Author):

The authors have addressed my concerns.

REVIEWERS' COMMENTS

Reviewer #1 (Remarks to the Author):

All my original concerns have been addressed

Reviewer #3 (Remarks to the Author):

The authors have addressed my concerns.

We thank the reviewers for their critiques of our work and believe the manuscript is much stronger by addressing their concerns.